# STUDENT SPECIALIZATION IN DEEP ReLU NETWORKS WITH FINITE WIDTH AND INPUT DIMENSION

## ABSTRACT

To analyze deep ReLU network, we adopt a student-teacher setting in which an over-parameterized student network learns from the output of a fixed teacher network of the same depth, with Stochastic Gradient Descent (SGD). Our contributions are two-fold. First, we prove that when the gradient is small at every training sample, student node *specializes* to teacher nodes in the lowest layer under mild conditions. Second, analysis of noisy recovery and training dynamics in 2-layer network shows that strong teacher nodes (with large fan-out weights) are learned first and subtle teacher nodes are left unlearned until late stage of training. As a result, it could take a long time to converge into these small-gradient critical points. Our analysis shows that over-parameterization is a necessary condition for specialization to happen at the critical points, and helps student nodes cover more teacher nodes with fewer iterations. Both improve generalization. Different from Neural Tangent Kernel (Jacot et al., 2018) and statistical mechanics approach (Goldt et al., 2019), our approach operates on finite width, mild over-parameterization (as long as there are more student nodes than teacher) and finite input dimension. Experiments justify our finding.

## 1 INTRODUCTION

Deep Learning has achieved great success in the recent years (Silver et al., 2016; He et al., 2016; Devlin et al., 2018). Although networks with even one-hidden layer can fit any function (Hornik et al., 1989), it remains an open question how such networks can generalize to new data. Different from what traditional machine learning theory predicts, empirical evidence (Zhang et al., 2017) shows more parameters in neural network lead to better generalization. How over-parameterization yields strong generalization is an important question for understanding how deep learning works.

In this paper, we analyze deep ReLU networks with *teacher-student* setting: a fixed teacher network provides the output for a student to learn via SGD. Both teacher and student are deep ReLU networks. Similar to (Goldt et al., 2019), the student is *over-realized* compared to the teacher: at each layer $l$, the number $n_l$ of student nodes is larger than the number $m_l$ of teacher ($n_l > m_l$). Although over-realization is different from *over-parameterization*, i.e., the total number of parameters in the student model is larger than the training set size $N$, over-realization directly correlates with the width of networks and is a measure of over-parameterization.

The student-teacher setting has a long history (Saad & Solla, 1996; 1995; Freeman & Saad, 1997; Mace & Coolen, 1998) and recently gains increasing interest (Goldt et al., 2019; Aubin et al., 2018) in analyzing 2-layered network. While worst-case performance on arbitrary data distributions may not be a good model for real structured dataset and can be hard to analyze, using a teacher network implicitly enforces an inductive bias and could potentially lead to better generalization bound.

Specialization, that is, a student node becomes increasingly correlated with a teacher node during training (Saad & Solla, 1996), is one of the important topic in this setup. If all student nodes are specialized to the teacher, then student tends to output the same as the teacher and generalization performance can be expected. Empirically, it has been observed in 2-layer networks (Saad & Solla, 1996; Goldt et al., 2019) and multi-layer networks (Tian et al., 2019; Li et al., 2016), in both synthetic and real dataset. In contrast, theoretical analysis is limited with strong assumptions (e.g., Gaussian inputs, infinite input dimension, local convergence, 2-layer setting, small number of hidden nodes). In this paper, with arbitrary training distribution and finite input dimension, we show rigorously that when gradient at each training sample is small (i.e., the interpolation setting as suggested in (Ma

et al., 2017; Liu & Belkin, 2018; Bassily et al., 2018)), the student node at the lowest layer can be proven to *specialize* to the teacher nodes: **each teacher node is aligned with at least one student node in the lowest layer**. This explains one-to-many mapping between teacher and student nodes and the existence of un-specialized student nodes, as observed empirically in (Saad & Solla, 1996). Furthermore, from the proof condition, more over-realization encourages specialization.

Our setting is different from previous works. **(1)** While statistical mechanics approaches (Saad & Solla, 1996; Goldt et al., 2019; Gardner & Derrida, 1989; Aubin et al., 2018) assume both the training set size $N$ and the input dimension $d$ goes to infinite (i.e., the thermodynamics limits) and assume Gaussian inputs, our analysis allows finite $d$ and impose *no* parametric constraints on the input data distribution. **(2)** While Neural Tangent Kernel (Jacot et al., 2018; Du et al., 2018b) and mean-field approaches (Mei et al., 2018) requires infinite (or very large) width, our setting applies to finite width as long as student is slightly over-realized ($n_l \geq m_l$). In this paper we study the infinite training sample case (the training set is a region), and leave finite sample analysis as the future work.

In addition, we further analyze the training dynamics and show that most student nodes converge first towards strong teacher nodes with large fan-out weights in magnitude. While this makes training robust to dataset noise and naturally explains implicit regularization, the same mechanism also leaves weak teacher nodes unexplained until very late stage of training, yielding high generalization error with finite iterations. In this situation, we show that over-realization plays another important role: once the strong teacher nodes have been covered, there are always spare student nodes ready to switch to weak teacher nodes quickly. Empirically, we show more teacher nodes are covered with the same number of iterations, and generalization is also improved.

We verify our findings with numerical experiments. Starting with 2-layer setting, we justify Theorem 2 and Theorem 3 with Gaussian inputs, showing one-to-many specialization and existence of un-specialized nodes. For deep ReLU networks, we show specialization happens not only in the lowest layer, as suggested by Theorem 4, but also in other hidden layers, on both Gaussian inputs and CIFAR10. We also perform ablation studies about the effect of student over-realization. For training dynamics, we show the strong/weak teacher effects in 2-layer settings and over-realization could improve specialization and generalization.

## 2 RELATED WORKS

**Student-teacher setting**. This setting has a long history (Engel & Van den Broeck, 2001; Gardner & Derrida, 1989). The seminar works (Saad & Solla, 1996; 1995) studies 1-hidden layer case from statistical mechanics point of view in which the input dimension goes to infinity, or so-called *thermodynamics limits*. They study symmetric solutions and locally analyze the symmetric breaking behavior and onset of *specialization* of the student nodes towards the teacher. Recent follow-up works (Goldt et al., 2019) makes the analysis rigorous and empirically shows that random initialization and training with SGD indeed gives student specialization in 1-hidden layer case, which is consistent with our experiments. With the same assumption, (Aubin et al., 2018) studies phase transition property of specialization in 2-layer networks with small number of hidden nodes using replica formula. In these works, inputs are assumed to be Gaussian and step or Gauss error function is used as nonlinearity. Few works study teacher-student setting with more than two layers. (Allen-Zhu et al., 2019a) shows the recovery results for 2 and 3 layer networks, with modified SGD, batchsize 1 and heavy over-parameterization.

In comparison, our work shows that specialization happens around the SGD critical points in the lowest layer for deep ReLU networks, without any parametric assumptions of input distribution.

**Local minima is Global**. While in deep linear network, all local minima are global (Laurent & Brecht, 2018; Kawaguchi, 2016), situations are quite complicated with nonlinear activations. While local minima is global when the network has invertible activation function and distinct training samples (Nguyen & Hein, 2017; Yun et al., 2018) or Leaky ReLU with linear separate input data (Laurent & von Brecht, 2017), multiple works (Du et al., 2018a; Ge et al., 2017; Safran & Shamir, 2017; Yun et al., 2019) show that in GD case with population or empirical loss, spurious local minima can happen even in two-layered network. Many are specific to two-layer and hard to generalize to multi-layer setting. In contrast, our work brings about a generic formulation for deep ReLU network and gives recovery properties in the student-teacher setting.

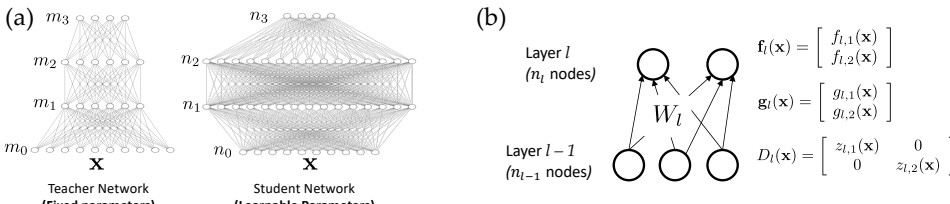

Figure 1: Problem Setup. **(a)** Student-teacher setting. The student network learns from the output of a fixed teacher network via stochastic gradient descent (SGD). **(b)** Notations. All low cases are scalar, bolds are column vectors and upper cases are matrices.

**Learning wild networks**. Recent works on Neural Tangent Kernel (Jacot et al., 2018; Du et al., 2018b; Allen-Zhu et al., 2019b) show the global convergence of GD for multi-layer networks with infinite width. (Li & Liang, 2018) shows the convergence in one-hidden layer ReLU network using GD/SGD to solution with good generalization, when the input data are assumed to be clustered into classes. Both lines of work assume heavily over-parameterized network, requiring polynomial growth of number of nodes with respect to the number of samples. (Chizat & Bach, 2018) shows global convergence of over-parameterized network with optimal transport. (Tian et al., 2019) assumes mild over-realization and gives convergence results for 2-layer network when a subset of the student network is close to the teacher. Our work extends it with much weaker assumptions.

**Deep Linear networks**. For deep linear networks, multiple works (Lampinen & Ganguli, 2019; Saxe et al., 2013; Arora et al., 2019; Advani & Saxe, 2017) have shown interesting training dynamics. One common assumption is that the singular spaces of weights at nearby layers are aligned at initialization, which decouples the training dynamics. Such a nice property would not hold for nonlinear network. (Lampinen & Ganguli, 2019) shows that under this assumption, weight components with large singular value are learned first, while we analyze and observe empirically similar behaviors on the student node level. Generalization property of linear networks can also be analyzed in the limit of infinite input dimension with teacher-student setting (Lampinen & Ganguli, 2019). However, deep linear networks lack specialization which plays a crucial role in the nonlinear case. To our knowledge, we are the first to analyze specialization rigorously in deep ReLU networks.

## 3 MATHEMATICAL FRAMEWORK

**Notation**. Consider a student network and its associated teacher network (Fig. 1(a)). Denote the input as $\mathbf{x}$. We focus on multi-layered networks with $\sigma(\cdot)$ as ReLU nonlinearity. We use the following equality extensively: $\sigma(x) = \mathbb{I}[x > 0]x$, where $\mathbb{I}[\cdot]$ is the indicator function. For node $j$, $f_j(\mathbf{x})$, $z_j(\mathbf{x})$ and $g_j(\mathbf{x})$ are its activation, gating function and backpropagated gradient *after the gating*.

Both teacher and student networks have $L$ layers. The input layer is layer 0 and the topmost layer (layer that is closest to the output) is layer $L$. For layer $l$, let $m_l$ be the number of teacher node while $n_l$ be the number of student node. The weights $W_l \in \mathbb{R}^{n_{l-1} \times n_l}$ refers to the weight matrix that connects layer $l-1$ to layer $l$ on the student side. $W_l = [\mathbf{w}_{l,1}, \mathbf{w}_{l,2}, \ldots, \mathbf{w}_{l,n_l}]$ where each $\mathbf{w} \in \mathbb{R}^{n_{l-1}}$ is the weight vector. Similarly we have teacher weight $W_l^* \in \mathbb{R}^{m_{l-1} \times m_l}$. Denote $\mathcal{W} = \{W_1, W_2, \ldots, W_L\}$ as the collection of all trainable parameters.

Let $\mathbf{f}_l(\mathbf{x}) = [f_{l,1}(\mathbf{x}), \ldots, f_{l,n_l}(\mathbf{x})]^\mathsf{T} \in \mathbb{R}^{n_l}$ be the activation vector of layer $l$, $D_l(\mathbf{x}) = \mathrm{diag}[z_{l,1}(\mathbf{x}), \ldots, z_{l,n_l}(\mathbf{x})] \in \mathbb{R}^{n_l \times n_l}$ be the diagonal matrix of gating function (for ReLU it is either 0 or 1), and $\mathbf{g}_l(\mathbf{x}) = [g_{l,1}(\mathbf{x}), \ldots, g_{l,n_l}(\mathbf{x})]^\mathsf{T} \in \mathbb{R}^{n_l}$ be the backpropated gradient vector. By definition, the input layer has $\mathbf{f}_0(\mathbf{x}) = \mathbf{x} \in \mathbb{R}^{n_0}$ and $m_0 = n_0$. Note that $\mathbf{f}_l(\mathbf{x})$, $\mathbf{g}_l(\mathbf{x})$ and $D_l(\mathbf{x})$ are all dependent on $\mathcal{W}$. For brevity, we often use $\mathbf{f}_l(\mathbf{x})$ rather than $\mathbf{f}_l(\mathbf{x}; \mathcal{W})$.

All notations with superscript $^*$ are from the teacher, only dependent on the teacher and remains the same throughout the training. $D_L^*(\mathbf{x}) = D_L(\mathbf{x}) \equiv I_{C \times C}$ since there is no ReLU gating. Note that $C$ is the dimension of output for both teacher and student. With the notation, gradient descent is:

$$\dot{W}_l = \mathbb{E}_\mathbf{x}\left[\mathbf{f}_{l-1}(\mathbf{x})\mathbf{g}_l^\mathsf{T}(\mathbf{x})\right] \tag{1}$$

In SGD, the expectation $\mathbb{E}_\mathbf{x}[\cdot]$ is taken over a batch. In GD, it is over the entire dataset.

*Bias term.* With the same notation we can also include the bias term. In this case, $W_l \in \mathbb{R}^{(n_{l-1}+1) \times n_l}$, $\mathbf{w}_{l,1} = [\tilde{\mathbf{w}}; b] \in \mathbb{R}^{n_{l-1}+1}$, $\mathbf{f}_l \in \mathbb{R}^{n_l+1}$ (last column is all one), $\mathbf{g}_l \in \mathbb{R}^{n_l+1}$ and $D_l \in \mathbb{R}^{(n_l+1) \times (n_l+1)}$ (last diagonal element is always 1).

**Objective**. We assume that both the teacher and the student output a vector. We use the output of teacher as the input of the student and the objective is:

$$\min_{\mathcal{W}} J(\mathcal{W}) = \frac{1}{2} \mathbb{E}_{\mathbf{x}} \left[ \|\mathbf{f}_L^*(\mathbf{x}) - \mathbf{f}_L(\mathbf{x})\|^2 \right] \tag{2}$$

We want to ask the following qeustion:

*Are student nodes specialized to teacher nodes at the same layers after training?*    (*)

One might wonder this is hard since the student's intermediate layer receives no *direct supervision* from the corresponding teacher layer, but relies only on backpropagated gradient. Surprisingly, the following theorem shows that it is possible for every intermediate layer:

**Lemma 1** (Recursive Gradient Rule). *At layer $l$, the backpropagated $\mathbf{g}_l(\mathbf{x})$ satisfies*

$$\mathbf{g}_l(\mathbf{x}) = D_l(\mathbf{x}) \left[ A_l(\mathbf{x}) \mathbf{f}_l^*(\mathbf{x}) - B_l(\mathbf{x}) \mathbf{f}_l(\mathbf{x}) \right], \tag{3}$$

*where the mixture coefficient $A_l(\mathbf{x}) = V_l^\mathsf{T}(\mathbf{x}) V_l^*(\mathbf{x}) \in \mathbb{R}^{n_l \times m_l}$ and $B_l(\mathbf{x}) = V_l^\mathsf{T}(\mathbf{x}) V_l(\mathbf{x}) \in \mathbb{R}^{n_l \times n_l}$. The matrices $V_l(\mathbf{x}) \in \mathbb{R}^{C \times n_l}$ and $V_l^*(\mathbf{x}) \in \mathbb{R}^{C \times m_l}$ are defined in a top-down manner:*

$$V_{l-1}(\mathbf{x}) = V_l(\mathbf{x}) D_l(\mathbf{x}) W_l^\mathsf{T}, \quad V_{l-1}^*(\mathbf{x}) = V_l^*(\mathbf{x}) D_l^*(\mathbf{x}) W_l^{*\mathsf{T}} \tag{4}$$

*In particular, $V_L(\mathbf{x}) = V_L^*(\mathbf{x}) = I_{C \times C}$.*

For convenience, we can write $V_l(\mathbf{x}) = [\mathbf{v}_{l,1}(\mathbf{x}), \mathbf{v}_{l,2}(\mathbf{x}), \dots, \mathbf{v}_{l,n_l}(\mathbf{x})]$, then we have each element of $A_l$, $\alpha_{l,jj'}(\mathbf{x}) = \mathbf{v}_{l,j}^\mathsf{T}(\mathbf{x}) \mathbf{v}_{l,j'}^*(\mathbf{x})$ and element of $B_l$, $\beta_{l,jj'}(\mathbf{x}) = \mathbf{v}_{l,j}^\mathsf{T}(\mathbf{x}) \mathbf{v}_{l,j'}(\mathbf{x})$. Note that Lemma 1 applies to arbitrarily deep ReLU networks and allows different number of nodes for the teacher and student. In particular, student can be over-parameterized (or over-realized).

Let $R_0 = \{\mathbf{x} : \rho(\mathbf{x}) > 0\}$ be the *infinite* training set, where $\rho(\mathbf{x})$ is the input data distribution. Let $R_l = \{\mathbf{f}_l(\mathbf{x}) : \mathbf{x} \in R_0\}$, which is the image of the training set at the output of layer $l$, and also a convex polytope. Then the mixture coefficient $A_l(\mathbf{x})$ and $B_l(\mathbf{x})$ have the following property:

**Corollary 1** (Piecewise constant). *$R_0$ can be decomposed into a finite (but potentially exponential) set of regions $\mathcal{R}_{l-1} = \{R_{l-1}^1, R_{l-1}^2, \dots, R_{l-1}^J\}$. $A_l(\mathbf{x})$ and $B_l(\mathbf{x})$ are constant in $R_{l-1}^j$.*

## 4   CRITICAL POINT ANALYSIS

We first show that due to property of ReLU node and subset sampling in SGD, at SGD critical point, under mild condition, the teacher node aligns with at least one student node and the goal (*) can be reached in the lowest layer.

**Definition 1** (SGD critical point). *$\hat{\mathcal{W}}$ is a SGD critical point if for any batch, $\dot{W}_l = 0$ for $1 \le l \le L$.*

**Theorem 1** (Interpolation). *Denote $\mathcal{D} = \{\mathbf{x}_i\}$ as a dataset of $N$ samples. If $\hat{\mathcal{W}}$ is a critical point for SGD, then either $\mathbf{g}_l(\mathbf{x}_i; \hat{\mathcal{W}}) = \mathbf{0}$ or $\mathbf{f}_{l-1}(\mathbf{x}_i; \hat{\mathcal{W}}) = \mathbf{0}$.*

Such critical points exist since over-realized student can mimic teacher perfectly. Note that critical points in SGD is much stronger than those in GD, where the gradient is always averaged at a fixed data distribution. If $\mathbf{f}_{l-1}$ has a bias term (and $\mathbf{f}_{l-1} \neq \mathbf{0}$ always), then $\mathbf{g}_l(\mathbf{x}_i; \hat{\mathcal{W}}) = \mathbf{0}$. For topmost layer, immediately we have $\mathbf{g}_L(\mathbf{x}_i; \hat{\mathcal{W}}) = \mathbf{f}_L^*(\mathbf{x}_i) - \mathbf{f}_L(\mathbf{x}_i) = \mathbf{0}$, which is global optimum with zero training loss. In the following, we want to check whether this condition leads to specialization, i.e., whether the teacher's weights are recovered/aligned by the student, i.e., whether for teacher $j$, there exists a student $k$ at the same layer so that $\mathbf{w}_k = \gamma \mathbf{w}_j$ for some $\gamma > 0$.

Note that $\mathbf{g}_l(\mathbf{x}_i; \hat{\mathcal{W}}) = \mathbf{0}$ might be a strong assumption since in practice the gradient is small but never zero. A weaker assumption is that $\|\mathbf{g}_l(\mathbf{x}_i; \hat{\mathcal{W}})\|_\infty \le \epsilon$ or even $\mathbb{E}_t \left[ \|\mathbf{g}_l(\mathbf{x}_i; \hat{\mathcal{W}})\|_\infty \right] \le \epsilon$. For this, Theorem 5 shows (approximate) alignment/specialization still holds for noisy case.

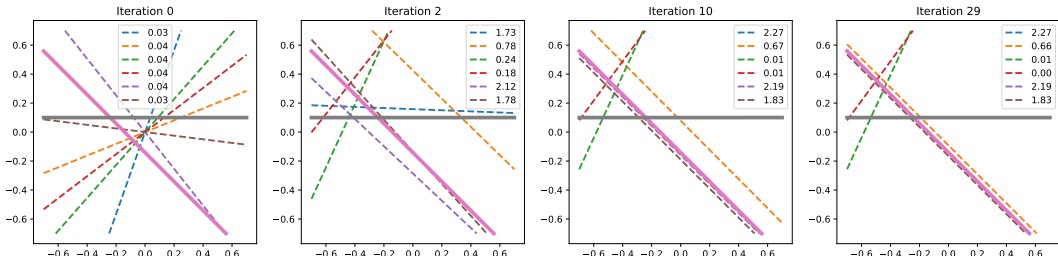

Figure 2: Convergence (2 dimension) for 2 teachers (solid line) and 6 students (dashed line). Legend shows $\|\mathbf{v}_k\|$ for student node $k$. $\|\mathbf{v}_k\| \to 0$ for nodes that are not aligned with teacher.

### 4.1  ASSUMPTION OF TEACHER NETWORK

Obviously, an arbitrary teacher network won't be reconstructed. A trivial example is that a teacher network always output $0$ since all the training samples lie in the inactive halfspace of its ReLU nodes. Therefore, we need to impose condition on the teacher network.

Let $E_j = \{\mathbf{x} : f_j(\mathbf{x}) > 0\}$ be the activation region of node $j$. Note that the halfspace $E_j$ is an open set. Let $\partial E_j = \{\mathbf{x} : f_j(\mathbf{x}) = 0\}$ be the decision boundary of node $j$.

**Definition 2** (Observer). *Node $k$ is an observer of node $j$ if $E_k \cap \partial E_j \neq \emptyset$.*

**Assumption 1** (Teacher Network). *For each layer $l$, we require that (1) the teacher weights $\mathbf{w}_{l,j}^*$ are not co-linear. and (2) the boundary of $\mathbf{w}_{l,j}^*$ is visible in the training set: $\partial E_{l,j}^* \cap R_{l-1} \neq \emptyset$.*

Assumption 1 is our assumption of the teacher. The first requirement is trivial. The second one is reasonable since two teacher nodes who behaves linearly in the training set are indistinguishable.

### 4.2  ALIGNMENT OF TEACHER WITH STUDENT, 2-LAYER CASE

We first start with 2-layer case, in which $A_1(\mathbf{x})$ and $B_1(\mathbf{x})$ are constant with respect to $\mathbf{x}$, since there is no ReLU gating at the top layer $l = 2$. In this case, from the SGD critical point at $l = 1$, $\mathbf{g}_1(\mathbf{x}) = D_1(\mathbf{x})\left[A_1 \mathbf{f}_1^*(\mathbf{x}) - B_1 \mathbf{f}_1(\mathbf{x})\right] = \mathbf{0}$, alignment between teacher and student can be achieved:

**Theorem 2** (Student-teacher Alignment, 2-layers). *With Assumption 1, at SGD critical point, if a teacher node $j$ is observed by a student node $k$ and $\alpha_{kj} \neq 0$, then there exists at least one student node $k'$ aligned with $j$.*

The intuition is that if the input $\mathbf{x}$ takes sufficiently diverse values, ReLU activations $\sigma(\mathbf{w}_k^\mathsf{T}\mathbf{x})$ can be proven to be mutually linear independent. On the other hand, the gradient of each student node $k$ *when active*, is $\boldsymbol{\alpha}_k^\mathsf{T} \mathbf{f}(\mathbf{x}) - \mathbf{b}_k^\mathsf{T} \mathbf{f}_1(\mathbf{x}) = 0$, a linear combination of teacher and student nodes (note $\boldsymbol{\alpha}_k^\mathsf{T}$ and $\boldsymbol{\beta}_k^\mathsf{T}$ are $k$-th rows of $A_1$ and $B_1$). Therefore, zero gradient means that the summation of coefficients of co-linear ReLU nodes is zero. Since teachers are not co-linear, any teacher node is co-linear with at least one student node. Alignment with multiple student nodes is also possible. If there is no nonlinearity (e.g., deep linear models), alignment won't happen since a linear subspace has many representations.

Note that a necessary condition of a reconstructed teacher node is that its boundary is in the active region of student, or is *observed* (Definition 2). This is intuitive since a teacher node which behaves like a linear node is partly indistinguishable from a bias term. This also suggests that over-parameterization (more student nodes) are important. More student nodes mean more observers, and the existence argument in Theorem 4 is more likely to happen and more teacher nodes can be covered by student, yielding better generalization.

For student nodes that are not aligned with the teacher, if they are observed by other student nodes, then following a similar logic, we have the following:

**Theorem 3** (Prunable Un-specialized Student Nodes). *With Assumption 1, at SGD critical point, if an unaligned student $k$ has $C$ independent observers (concatenating $\mathbf{v}$ yields a full rank matrix), then $\sum_{k' \in \text{co-linear}(k)} \mathbf{v}_{k'} \|\mathbf{w}_{k'}\| = \mathbf{0}$. If node $k$ is not co-linear with any other student, then $\mathbf{v}_k = \mathbf{0}$.*

**Corollary 2.** *With sufficient observers, the contribution of all unaligned student nodes is zero.*

Theorem 3 and Corollary 2 open the way of network pruning (LeCun et al., 1990; Hassibi et al., 1993; Hu et al., 2016). This is consistent with Theorem 5 in (Tian et al., 2019) which also shows the fan-out weights are zero up on convergence in 2-layer networks, if the initialization is close. In contrast, Theorem 3 analyzes the critical point rather than the dynamics.

Note that a relate theorem (Theorem 6) in (Laurent & von Brecht, 2017) studies 2-layer network with scalar output and linear separable input, and discusses characteristics of individual data point contributing loss in a local minima of GD. Here no linear separable condition is imposed.

### 4.3 MULTI-LAYER CASE

Thanks to Lemma 1 which holds for deep ReLU networks, we can use similar intuition to analyze the behavior of the lowest layer ($l = 1$) in the multiple layer case. The difference here is that $A_1(\mathbf{x})$ and $B_1(\mathbf{x})$ are no longer constant over $\mathbf{x}$. Fortunately, using Corollary 1, we know that $A_1(\mathbf{x})$ and $B_1(\mathbf{x})$ are piece-wise constant that separate the input region $R_0$ into a finite (but potentially exponential) set of constant regions $\mathcal{R}_0 = \{R_0^1, R_0^2, \ldots, R_0^J\}$ plus a zero-measure set. This suggests that we could check each region separately. If the boundary of a teacher $j$ and a student $k$ lies in the region, similar logic applies (here $\alpha_{kj}$ is the $(k, j)$ entry of $A_1(\mathbf{x})$ and is constant in a region $R \in \mathcal{R}_0$).

**Theorem 4** (Student-teacher Alignment, Multiple Layers). *With Assumption 1, at SGD critical points, for any teacher node $j$ at $l = 1$, if there exists a region $R \in \mathcal{R}$ and a student observer $k$ so that $\partial E_j^* \cap E_k \cap R \neq \emptyset$ and $\alpha_{kj}(R) \neq 0$, then node $j$ aligns with at least one student node $k'$.*

Note that even with random $V_1(\mathbf{x})$ (e.g., at initialization), Theorem 4 still holds with high probability (when $\alpha_{kj} \neq 0$) and teacher $\mathbf{f}_1^*(\mathbf{x})$ can still align with student $\mathbf{f}_1(\mathbf{x})$. This suggests a picture of *bottom-up training* in backpropagation: After the alignment of activations at layer 1, we just treat layer 1 as the low-level features and the procedure repeats until the student matches with the teacher at all layers. This is consistent with many previous works that empirically show the network is learned in a bottom-up manner (Li et al., 2018).

Note that the alignment may happen concurrently across layers: if the activations of layer 1 start to align, then activations of layer 2, which depends on activations of layer 1, will also start to align since there now exists a $W_2$ that yields strong alignments, and so on. This creates a *critical path* from important student nodes at the lowest layer all the way to the output, and this critical path accelerates the convergence of that student node. We leave a formal analysis to the future work.

**Small Gradient Case**. In practice, stochastic gradient (or its expectation over time) fluctuates around zero ($\|\mathbf{g}_1(\mathbf{x})\|_\infty \leq \epsilon$, or $\mathbb{E}_t [\|\mathbf{g}_1(\mathbf{x})\|_\infty] \leq \epsilon$), but never zero. In this case, Theorem 5 shows that a rough specialization still follows. The ratio of recovery is also shown for weights/biases separately, as a function of $\epsilon$. Note $\tilde{\theta}_{jj'}$ is the angle of two weights $\tilde{\mathbf{w}}_j$ and $\tilde{\mathbf{w}}_{j'}$.

**Theorem 5** (Noisy Recovery). *If Assumption 1 holds and any two teachers $\mathbf{w}_j^*$, $\mathbf{w}_{j'}^*$ satisfy $\tilde{\theta}_{jj'} \geq \theta_0 > 0$ or $|b_{j'}^* - b_j^*| \geq b_0 > 0$. Suppose $\|\mathbf{g}_1(\mathbf{x}, \hat{\mathcal{W}})\|_\infty \leq \epsilon$ for any $\mathbf{x} \in R_0$ with $\epsilon \leq \epsilon_0$, then for any teacher $j$ at $l = 1$, if there exists a region $R \in \mathcal{R}$ and a student observer $k$ so that $\partial E_j^* \cap E_k \cap R \neq \emptyset$, and $\alpha_{kj}(R) \neq 0$, then $j$ is roughly aligned with a student $k'$: $\sin \theta_{jk'} = \mathcal{O}\left(\frac{\epsilon^{1-\delta}}{|\alpha_{kj}|}\right)$ and $|b_j^* - b_{k'}| = \mathcal{O}\left(\frac{\epsilon^{1-2\delta}}{|\alpha_{kj}|}\right)$ for any $\delta > 0$. The hidden constants depends on $\delta$, $\epsilon_0$ and the size of region $\partial E_j^* \cap E_k \cap R$.*

Note that $\mathbb{E}_t [\|\mathbf{g}_1(\mathbf{x})\|_\infty] \leq \epsilon$ leads to $\|\mathbf{g}_1(\mathbf{x})\|_\infty \leq \epsilon$ at least for some iteration $t$. Therefore, Theorem 5 still applies since it does not rely on past history of the weight/gradient. Note that Theorem 5 assumes infinite number of data points and leave finite sample case to future work.

## 5 ANALYSIS ON TRAINING DYNAMICS

Our analysis so far shows student specialization happens at SGD critical points under mild conditions. A natural question arises: is running SGD long enough sufficient to achieve these critical points? Some previous works (Ge et al., 2017; Livni et al., 2014) show that empirically SGD does not recover the parameters of a teacher network up to permutation, while other works (Saad & Solla, 1996; Goldt et al., 2019) show specialization happens. Why there is a discrepancy? There are several reasons. First, from Theorem 3, there exist un-specialized student nodes, so a simple permutation test on student weights might fail. Second, as suggested by Theorem 5, it can take a long time to

recover a teacher node $k$ with small $\|\mathbf{v}_k^*\|$ (since $\alpha_{kj} = \mathbf{v}_k^{*\mathsf{T}}\mathbf{v}_j$). In fact, if $\mathbf{v}_k^* = \mathbf{0}$ then it has no contribution to the output and recovery never happens. This is particularly problematic if the output dimension is 1 (scalar output), since a single small teacher weight $v_k^*$ would block the recovery of the entire teacher node $k$. Previous works (Lampinen & Ganguli, 2019) shows similar behaviors in the dynamics of singular values in deep linear networks in teacher-student setting, which lack student specialization. Here we study these behaviors in deep ReLU networks.

In the following, we analyze various local dynamic behaviors of 2-layer ReLU network. Due to the complexity, we leave a formal characterization of the entire training procedure for future work.

**Definition 3.** *A teacher node $j$ is strong (or weak), if $\|\mathbf{v}_j^*\|$ is large (or small).*

In this case, the dynamics can be written as the following:

$$\dot{\mathbf{w}}_k = \mathbb{E}_{\mathbf{x}}\left[\mathbf{f}_{l-1}z_k[\mathbf{f}_l^{*\mathsf{T}}\boldsymbol{\alpha}_k - \mathbf{f}_l^\mathsf{T}\boldsymbol{\beta}_k]\right] = \mathbb{E}_{\mathbf{x}}\left[\mathbf{f}_{l-1}z_k[V_l^*\mathbf{f}_l^* - V_l\mathbf{f}_l]^\mathsf{T}\mathbf{v}_k\right] = \mathbb{E}_{\mathbf{x}}\left[\mathbf{f}_{l-1}z_k\mathbf{r}^\mathsf{T}\mathbf{v}_k\right], \quad (5)$$

where $V_1$ and $V_1^*$ are constant, $\boldsymbol{\alpha}_k = V_l^{*\mathsf{T}}\mathbf{v}_k$, $\boldsymbol{\beta}_k = V_l^\mathsf{T}\mathbf{v}_k$ and residue $\mathbf{r}_l = V_l^*\mathbf{f}_l^* - V_l\mathbf{f}_l \in \mathbb{R}^C$.

## 5.1 Weight magnitude

From Eqn. 5, we know that for both ReLU and linear network (since $f_k(\mathbf{x}) = z_k(\mathbf{x})\mathbf{w}_k^\mathsf{T}\mathbf{f}_{l-1}(\mathbf{x})$):

$$\frac{1}{2}\frac{\mathrm{d}\|\mathbf{w}_k\|^2}{\mathrm{d}t} = \mathbf{w}_k^\mathsf{T}\dot{\mathbf{w}}_k = \mathbb{E}_{\mathbf{x}}\left[f_k\mathbf{r}_l^\mathsf{T}\mathbf{v}_k\right] \quad (6)$$

When there is only a single output, $\mathbf{r}_l$ is a scalar and Eqn. 6 is simply an inner product between the residue and the activation of node $k$, over the batch. So if the node $k$ has activation which aligns well with the residual, the inner product is larger and $\|\mathbf{w}_k\|$ grows faster.

## 5.2 Angles between Teacher and Student weights

Note that Eqn. 6 only tell that the weight norm would increase, but didn't tell whether $\mathbf{w}_k$ converges to any teacher node $\mathbf{w}_j^*$. It could be the case that $\|\mathbf{w}_k\|$ goes up but doesn't move towards the teacher. To see that, let's check the quantity:

$$\mathbb{E}_{\mathbf{x}}\left[\mathbf{f}_{l-1}z_k f_j^*\right] = \mathbb{E}_{\mathbf{x}}\left[\mathbf{f}_{l-1}z_k z_j^*\mathbf{f}_{l-1}^\mathsf{T}\right]\mathbf{w}_j^* = G_{kj}\mathbf{w}_j^* \quad (7)$$

where $G_{kj} = \mathbb{E}_{\mathbf{x}}\left[\mathbf{f}_{l-1}z_k z_j^*\mathbf{f}_{l-1}^\mathsf{T}\right]$. Putting it in another way, we want to check the spectrum property of the PSD matrix $G_{kj}$. Intuitively, the direction of $\mathbb{E}_{\mathbf{x}}\left[\mathbf{f}_{l-1}z_k f_j^*\right]$ should lie between $\mathbf{w}_k$ and $\mathbf{w}_j^*$, and the magnitude is large when $\mathbf{w}_k$ and $\mathbf{w}_j^*$ are close to each other. This means that if $\mathbf{r}$ is dominated by a teacher $j$ (i.e., $\|\mathbf{v}_j^*\|$ is large), then $\dot{\mathbf{w}}_k$ would push $\mathbf{w}_k$ towards $\mathbf{w}_j^*$. This also shows that SGD will first try fitting strong teacher nodes, then weak teacher nodes.

Theorem 6 confirms this intuition if $\mathbf{f}_{l-1}$ follows spherical symmetric distribution (e.g., $\mathcal{N}(0, I)$).

**Theorem 6.** *If $\mathbf{f}_{l-1}$ follows spherical symmetric distribution, then $\mathbb{E}_{\mathbf{x}}\left[\mathbf{f}_{l-1}z_k f_j^*\right] \propto \frac{\|\mathbf{w}_j^*\|\|\mathbf{w}_k\|}{2}\left[(\pi - \theta)\mathbf{w}_j^* + \sin\theta\mathbf{w}_k\right]$, where $\theta$ is the angle between $\mathbf{w}_j^*$ and $\mathbf{w}_k$.*

As a result, for all $\theta \in [0, \pi]$, $\mathbb{E}_{\mathbf{x}}\left[\mathbf{f}_{l-1}z_k f_j^*\right]$ is always between $\mathbf{w}_j^*$ and $\mathbf{w}_k$ since $\pi - \theta$ and $\sin\theta$ are always non-negative. Without such symmetry, we assume the following holds:

**Assumption 2.** $\mathbb{E}_{\mathbf{x}}\left[\mathbf{f}_{l-1}z_k f_j\right] = \psi(\theta_{jk})\mathbf{w}_j + \psi'(\theta_{jk})\mathbf{w}_k$, *where $\psi(\pi) = 0$.*

Note that critical point analysis is applicable to any batch size, including 1. On the other hand, Assumption 2 holds when a moderately large batchsize leads to a decent estimation of the terms.

With this assumption, we can write the dynamics as $\dot{\mathbf{w}}_k = \|\mathbf{w}_k\|\mathbf{r}_k$, where the time-varying residue $\mathbf{r}_k$ of node $k$ is defined as the following ($\nu$ is a scalar related to $\psi'$):

$$\mathbf{r}_k = \sum_j \alpha_{jk}\psi(\theta_{jk})\mathbf{w}_j^* - \sum_{k'}\beta_{k'k}\psi(\theta_{k'k})\mathbf{w}_{k'} - \nu\mathbf{w}_k \quad (8)$$

## 5.3 Symmetric breaking, Winners-take-all and Focus Shifting

We could show that for two nodes $k \neq k'$, regardless of the form of $\mathbf{r}_k$, we have (note that $\bar{\mathbf{w}}$ is the length-normalized version of $\mathbf{w}$):

Figure 3: Student specialization of a 2-layered network with 10 teacher nodes and 1x/2x/5x/10x student nodes. $p$ is teacher polarity factor (Eqn. 9). For a student node $k$, we plot its normalized correlation (in terms of activation vector evaluated in a separate evaluation set) to its best correlated teacher as the $x$ coordinate and the fan-out weight norm $\|\mathbf{v}_k\|$ as the $y$ coordinate. We plot results from 32 random seed. Student nodes of different seeds are in different color. An un-specialized student node has low fan-out weight norm (Theorem 3).

.

**Theorem 7.** *For dynamics* $\dot{\mathbf{w}}_k = \|\mathbf{w}_k\|\mathbf{r}_k$, *we have* $\frac{\mathrm{d}}{\mathrm{d}t}\ln\frac{\|\mathbf{w}_k\|}{\|\mathbf{w}_{k'}\|} = \bar{\mathbf{w}}_k^\mathsf{T}\mathbf{r}_k - \bar{\mathbf{w}}_{k'}^\mathsf{T}\mathbf{r}_{k'}$.

We consider a special (and symmetric) case: $\mathbf{r}_k = \mathbf{r} = \mathbf{w}^* - \sum_k a_k\mathbf{w}_k$ with all $a_k > 0$, where $\mathbf{w}^*$ is a joint contribution of all teacher nodes. In this case, we could show that when $\bar{\mathbf{w}}_k^\mathsf{T}\mathbf{r}_k > \bar{\mathbf{w}}_{k'}^\mathsf{T}\mathbf{r}_{k'}$, $\frac{\mathrm{d}}{\mathrm{d}t}(\bar{\mathbf{w}}_k^\mathsf{T}\mathbf{r}_k - \bar{\mathbf{w}}_{k'}^\mathsf{T}\mathbf{r}_{k'}) < 0$ and vice versa. So the system provides negative feedback until $\bar{\mathbf{w}}_k = \bar{\mathbf{w}}_{k'}$ and according to Eqn. 7, the ratio between $\|\mathbf{w}_k\|$ and $\|\mathbf{w}_{k'}\|$ remains constant, after initial transition. We can also show that $\bar{\mathbf{w}}_k$ will align with $\mathbf{w}^*$ and every student node goes to $\mathbf{w}^*$.

However, due to Theorem 6, the net effect $\mathbf{w}^*$ can be *different* for different students and thus $\mathbf{r}_k$ are different. This opens the door for complicated dynamic behavior of neural network training.

**Symmetry breaking**. As one example, if we add a very small delta to some node, say $k = 1$ so that $\mathbf{r}_1 = \mathbf{r} + \epsilon\mathbf{w}^*$. Then to make $\frac{\mathrm{d}}{\mathrm{d}t}(\bar{\mathbf{w}}_k^\mathsf{T}\mathbf{r}_k - \bar{\mathbf{w}}_{k'}^\mathsf{T}\mathbf{r}_{k'}) = 0$, we have $\bar{\mathbf{w}}_k^\mathsf{T}\mathbf{r}_k > \bar{\mathbf{w}}_{k'}^\mathsf{T}\mathbf{r}_{k'}$ and thus according to Theorem 7, $\|\mathbf{w}_k\|/\|\mathbf{w}_{k'}\|$ grows exponentially. This symmetric breaking behavior provides a potential *winners-take-all* mechanism, since according to Theorem 6, the coefficient of $\mathbf{w}^*$ depends critically on the initial angle between $\mathbf{w}_k$ and $\mathbf{w}^*$.

**Strong teacher nodes are learned first**. If $\|\mathbf{v}_j^*\|$ is the largest among teacher nodes, then the joint $\mathbf{w}^*$ heavily biases towards teacher $j$ and all student nodes move towards teacher $j$. As a result, strong teacher learns first and is often covered by multiple co-linear students (Fig. 6, teacher-0).

**Focus shifting to weak teacher nodes**. The process above continues until residual along the direction of $\mathbf{w}_j^*$ quickly shrinks and residual corresponding to other teacher node (e.g., $\mathbf{w}_{j'}^*$ for $j' \neq j$) becomes dominant. Since each $\mathbf{r}_k$ is different, student node $k$ whose direction is closer to $\mathbf{w}_{j'}^*$ ($j' \neq j$) will shift their focus towards $\mathbf{w}_{j'}^*$, as shown in the green (shift to teacher-2) and magenta (shift to teacher-5) curves in Fig. 6.

**Possible slow convergence to weak teacher nodes**. While expected angle between two weights from initialization is $\pi/2$, shifting a student node $\mathbf{w}_k$ from chasing after a strong teacher node $\mathbf{w}_j^*$ to a weaker one $\mathbf{w}_{j'}^*$ could yield a large initial angle (e.g., close to $\pi$) between $\mathbf{w}_k$ and $\mathbf{w}_{j'}$. For example, all student nodes have been attracted to the opposite direction of a weak teacher node. In this case, the convergence can be arbitrarily slow. In fact, if there is only one teacher node and $\theta$ is the angle between teacher and student, then from Eqn. 8 we arrive at $\dot{\theta} \propto -\psi(\theta)\sin\theta$. Since $\psi(\theta)\sin\theta \sim (\pi - \theta)^2$ around $\theta = \pi$, the time spent from $\theta = \pi - \epsilon$ to some $\theta_0$ is $t_0 \sim \frac{1}{\epsilon} - \frac{1}{\pi - \theta_0} \to +\infty$ when $\epsilon \to 0$. In this case, over-realization helps by having more student nodes that are possibly ready for shifting towards weaker teachers, and thus accelerate convergence (Fig. 7). Alternatively, we could reinitialize those student nodes (Prakash et al., 2019).

## 6 EXPERIMENTS

We first verify our theoretical finding on synthetic dataset. We generate the input using $\mathcal{N}(0, \sigma^2 I)$ with $\sigma = 10$ and we sample 10k as training and another 10k as evaluation. For deep ReLU networks, we *regenerate* the dataset after every epoch to mimic infinite sample setting. The details of teacher/student construction is in Appendix (Sec. 8.16). The normalized correlation between nodes is computed in terms of activation vectors evaluated on a separate evaluation set.

**Two layer networks**. First we verify Theorem 2 and Theorem 3 in the 2-layer setting. Fig. 6 shows student nodes correlate with different teacher nodes over time. Fig. 3 shows for different degrees of

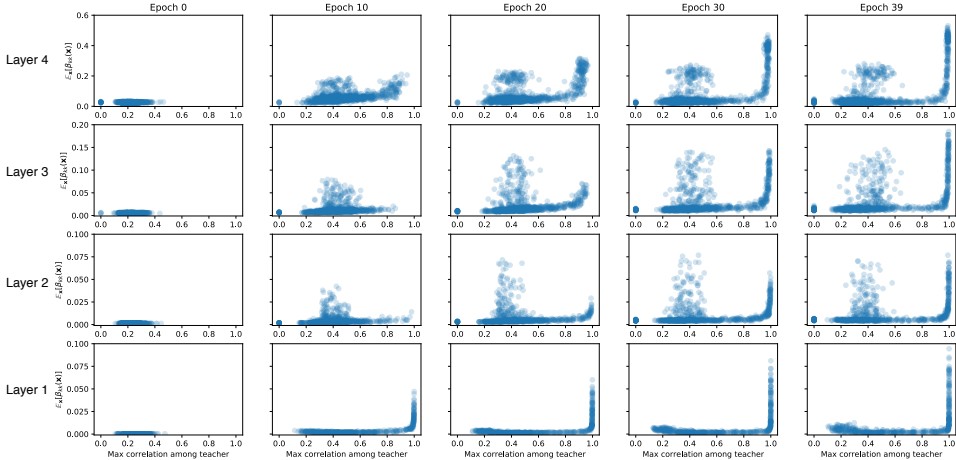

**Figure 4:** The strength of student specialization versus their fan-out coefficients in 4 layer ReLU network. Number of hidden teacher nodes is 50-75-100-125. Student is 10x over-realized. The dataset is *regenerated* with the input distribution after every epoch. For node $k$, y-axis is $\sqrt{\mathbb{E}_{\mathbf{x}}\left[\beta_{kk}(\mathbf{x})\right]}$, equivalent to the fan-out weight norm $\|\mathbf{v}_k\|$ in 2-layer case, and x-axis is its max correlation to the teachers. The lower layer learns first.

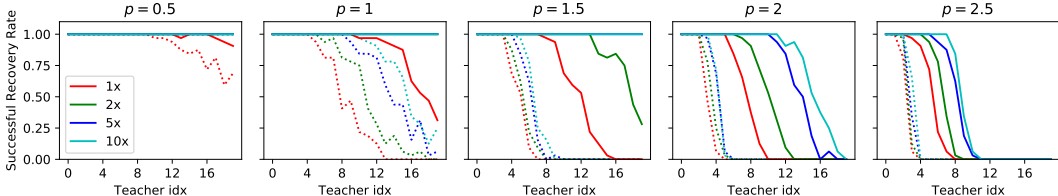

**Figure 5:** Success rate (over 32 trials with different random seeds) of recovery of 20 teacher nodes on 2-layer network at different teacher polarity $p$ (Eqn. 9) and different over-realization. Dotted line: successful rate after 5 epochs. Solid line: successful rate after 100 epochs.

over-realization ($1\times/2\times/5\times/10\times$), for nodes with weak specialization (i.e., its normalized correlation to the most correlated teacher is low), their magnitudes of fan-out weights are small. Otherwise the nodes with strong specialization have high fan-out weights.

**Deep Networks**. For deep ReLU networks, we observe specialization not only at the lowest layer, as suggested by Theorem 4, but also at multiple hidden layers. This is shown in Fig. 4. For each student node $k$, the x-axis is its best normalized correlation to teacher nodes, and y-axis is $\sqrt{\mathbb{E}_{\mathbf{x}}\left[\beta_{kk}(\mathbf{x})\right]}$, which is equivalent to $\|\mathbf{v}_k\|$ in 2-layer case. In the plot, we can also see the lowest layer learns first (the "L-shape" curve was established at epoch 10), then the top layers follow.

**Ablation on the effect of over-realization**. To further understand the role of over-realization, we plot the average rate of a teacher node that is matched with at least one student node successfully (i.e., correlation $> 0.95$). Fig. 5 shows that stronger teacher nodes are more likely to be matched, while weaker ones may not be explained well, in particular when the strength of the teacher nodes are polarized ($p$ is large). Over-realized student can explain more teacher nodes, while a student with $1\times$ nodes has sufficient capacity to fit the teacher perfectly, it gets stuck despite long training.

In addition, the evaluation loss (Appendix Fig. 11) shows that over-realization yields better generalization, in particular with large teacher node polarity ($p$ is large), where weak teacher nodes are hard to capture. For good performance on real datasets, getting weak teacher nodes can be important.

**Training Dynamics**. We set up a diverse strength of teacher node by constructing the fanout weights of teacher node $j$ as follows:

$$\|\mathbf{v}_j^*\| \sim 1/j^p, \tag{9}$$

where $p$ is the *teacher polarity factor* that controls how strong the energy decays across different teacher nodes. $p = 0$ means all teacher nodes are symmetric, and large $p$ means that the strength of teacher nodes are more polarized.

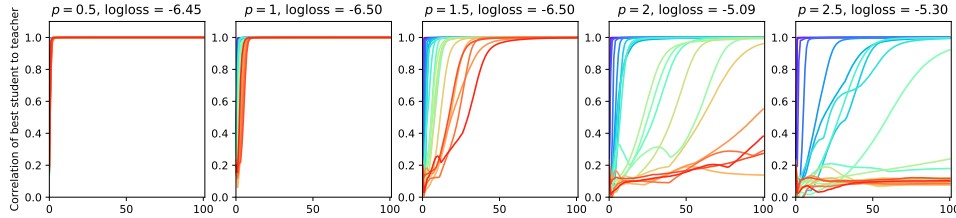

Figure 6: Student specialization with teacher polarity $p = 1$ (Eqn. 9). Same students are represented by the same color across plots. Three rows represent three different random seeds. We can see more students nodes specialize to `teacher-1` first. In contrast, `teacher-5` was not specialized until later by a node (e.g., magenta in the first row) that first chases after `teacher-1` then shifts its focus.

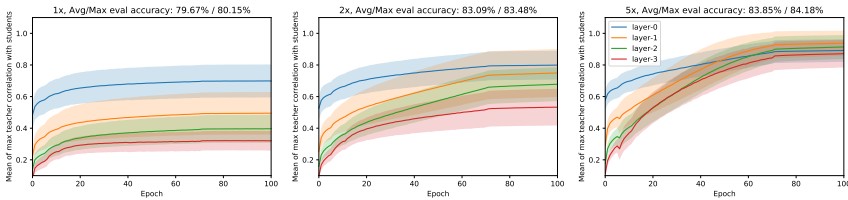

Figure 7: Evolution of best student correlation to teacher over iterations. Each rainbow color represents one of the 20 teachers (blue: strongest, red: weakest). 5x over-parameterization on 2-layer network.

Fig. 6 and Fig. 7 show that many student nodes specialize to a strong teacher node first. Once the strong teacher node was covered well, weaker teacher nodes are covered after many epochs.

**CIFAR-10**. We also experiment on CIFAR-10. We first pre-train a teacher network with 64-64-64-64 ConvNet (64 are channel sizes of the hidden layers, $L = 5$) on CIFAR-10 training set. Then the teacher network is pruned in a structured manner to keep strong teacher nodes. The student is over-realized based on teacher's remaining channels.

The convergence and specialization behaviors of student network is shown in Fig. 8. Specialization happens at all layers for different degree of over-realization. Over-realization boosts student specialization, measured by mean of maximal normalized correlation $\rho_{\mathrm{mean}} = \mathrm{mean}_{j \in \text{ teacher}} \max_{j' \in \text{ student}} \tilde{\mathbf{f}}_j^{* \top} \tilde{\mathbf{f}}_{j'}$ at each layer ($\tilde{\mathbf{f}}_j$ is the normalized activation of node $j$ over $N$ evaluation samples), and improves generalization, evaluated on CIFAR-10 evaluation set.

# 7 CONCLUSION AND FUTURE WORK

In this paper, we use student-teacher setting to analyze how an (over-parameterized) deep ReLU student network trained with SGD learns from the output of a teacher. When the magnitude of gradient per sample is small (student weights are near the critical points), the teacher can be proven to be covered by (possibly multiple) students and thus the teacher network is recovered in the lowest layer. By analyzing training dynamics, we also show that strong teacher node with large $\|\mathbf{v}^*\|$ is reconstructed first, while weak teacher node is reconstructed slowly. This reveals one important reason why the training takes long to reconstruct all teacher weights and why generalization improves with more training. As the next step, we would like to extend our analysis to finite sample case, and analyze the training dynamics in a more formal way. Verifying the insights from theoretical analysis on a large dataset (e.g., ImageNet) is also the next step.

Figure 8: Mean of the max teacher correlation $\rho_{\mathrm{mean}}$ with student nodes over epochs in CIFAR10. More over-realization gives better student specialization across all layers and achieves strong generalization (higher evaluation accuracy on CIFAR-10 evaluation set).

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

## 8   APPENDIX

### 8.1   LEMMA 1

*Proof.* We prove by induction. When $l = L$ we know that $\mathbf{g}_L(\mathbf{x}) = \mathbf{f}_L^*(\mathbf{x}) - \mathbf{f}_L(\mathbf{x})$, by setting $V_L^*(\mathbf{x}) = V_L(\mathbf{x}) = I_{C \times C}$ and the fact that $D_L(\mathbf{x}) = I_{C \times C}$ (no ReLU gating in the last layer), the condition holds.

Now suppose for layer $l$, we have:

$$
\begin{aligned}
\mathbf{g}_l(\mathbf{x}) &= D_l(\mathbf{x}) \left[ A_l(\mathbf{x})\mathbf{f}_l^*(\mathbf{x}) - B_l(\mathbf{x})\mathbf{f}_l(\mathbf{x}) \right] &(10)\\
&= D_l(\mathbf{x})V_l^\mathsf{T}(\mathbf{x}) \left[ V_l^*(\mathbf{x})\mathbf{f}_l^*(\mathbf{x}) - V_l(\mathbf{x})\mathbf{f}_l(\mathbf{x}) \right] &(11)
\end{aligned}
$$

Using

$$
\begin{aligned}
\mathbf{f}_l(\mathbf{x}) &= D_l(\mathbf{x})W_l^\mathsf{T}\mathbf{f}_{l-1}(\mathbf{x}) &(12)\\
\mathbf{f}_l^*(\mathbf{x}) &= D_l^*(\mathbf{x})W_l^{*\mathsf{T}}\mathbf{f}_{l-1}^*(\mathbf{x}) &(13)\\
\mathbf{g}_{l-1}(\mathbf{x}) &= D_{l-1}(\mathbf{x})W_l\mathbf{g}_l(\mathbf{x}) &(14)
\end{aligned}
$$

we have:

$$
\begin{aligned}
\mathbf{g}_{l-1}(\mathbf{x}) &= D_{l-1}(\mathbf{x})W_l\mathbf{g}_l(\mathbf{x}) &(15)\\
&= D_{l-1}(\mathbf{x}) \underbrace{W_l D_l(\mathbf{x})V_l^\mathsf{T}(\mathbf{x})}_{V_{l-1}^\mathsf{T}(\mathbf{x})} \left[ V_l^*(\mathbf{x})\mathbf{f}_l^*(\mathbf{x}) - V_l(\mathbf{x})\mathbf{f}_l(\mathbf{x}) \right] &(16)\\
&= D_{l-1}(\mathbf{x})V_{l-1}^\mathsf{T}(\mathbf{x}) \left[ \underbrace{V_l^*(\mathbf{x})D_l^*(\mathbf{x})W_l^{*\mathsf{T}}}_{V_{l-1}^*(\mathbf{x})} \mathbf{f}_{l-1}^*(\mathbf{x}) - \underbrace{V_l(\mathbf{x})D_l(\mathbf{x})W_l^\mathsf{T}}_{V_{l-1}(\mathbf{x})} \mathbf{f}_{l-1}(\mathbf{x}) \right] &(17)\\
&= D_{l-1}(\mathbf{x})V_{l-1}^\mathsf{T}(\mathbf{x}) \left[ V_{l-1}^*(\mathbf{x})\mathbf{f}_{l-1}^*(\mathbf{x}) - V_{l-1}(\mathbf{x})\mathbf{f}_{l-1}(\mathbf{x}) \right] &(18)\\
&= D_{l-1}(\mathbf{x}) \left[ A_{l-1}(\mathbf{x})\mathbf{f}_{l-1}^*(\mathbf{x}) - B_{l-1}(\mathbf{x})\mathbf{f}_{l-1}(\mathbf{x}) \right] &(19)
\end{aligned}
$$

$\square$

### 8.2   THEOREM 1

*Proof.* By definition of SGD critical point, we know that for any batch $\mathcal{B}_j$, Eqn. 1 vanishes:

$$
\dot{W}_l = \mathbb{E}_\mathbf{x} \left[ \mathbf{g}_l(\mathbf{x}; \hat{\mathcal{W}})\mathbf{f}_{l-1}^\mathsf{T}(\mathbf{x}; \hat{\mathcal{W}}) \right] = \sum_{i \in \mathcal{B}_j} \mathbf{g}_l(\mathbf{x}_i; \hat{\mathcal{W}})\mathbf{f}_{l-1}^\mathsf{T}(\mathbf{x}_i; \hat{\mathcal{W}}) = \sum_{i \in \mathcal{B}_j} U_i = 0 \qquad (20)
$$

where $U_i = \mathbf{g}_l(\mathbf{x}_i; \hat{\mathcal{W}})\mathbf{f}_{l-1}^\mathsf{T}(\mathbf{x}_i; \hat{\mathcal{W}})$. Note that $\mathcal{B}_j$ can be any subset of samples from the data distribution. Therefore, for a dataset of size $N$, Eqn. 20 holds for all $\binom{N}{|\mathcal{B}|}$ batches, but there are only $N$ data samples. With simple Gaussian elimination we know that for any $i_1 \neq i_2$, $U_{i_1} = U_{i_2} = U$. Plug that into Eqn. 20 we know $U = 0$ and thus for any $i$, $U_i = 0$. Since $U_i$ is an outer product, the theorem follows.

Note that if $\|\dot{W}_l\|_\infty \leq \epsilon$, which is $\| \sum_{i \in \mathcal{B}_j} U_i \|_\infty \leq \epsilon$, then with simple Gaussian elimination for two batches $\mathcal{B}_1$ and $\mathcal{B}_2$ with only two sample difference, we will have for any $i_1 \neq i_2$, $\|U_{i_1} - U_{i_2}\|_\infty = \| \sum_{i \in \mathcal{B}_1} U_i - \sum_{i \in \mathcal{B}_2} U_i \|_\infty \leq \| \sum_{i \in \mathcal{B}_1} U_i \|_\infty + \| \sum_{i \in \mathcal{B}_2} U_i \|_\infty = 2\epsilon$. Plug things back in and we have $|\mathcal{B}|\|U_i\|_\infty \leq [2(|\mathcal{B}| - 1) + 1]\epsilon$, which is $\|U_i\|_\infty \leq 2\epsilon$. If $\mathbf{f}_{l-1}(\mathbf{x}; \hat{\mathcal{W}})$ has the bias term, then immediately we have $\|\mathbf{g}_l(\mathbf{x}; \hat{\mathcal{W}})\|_\infty \leq \epsilon$. $\square$

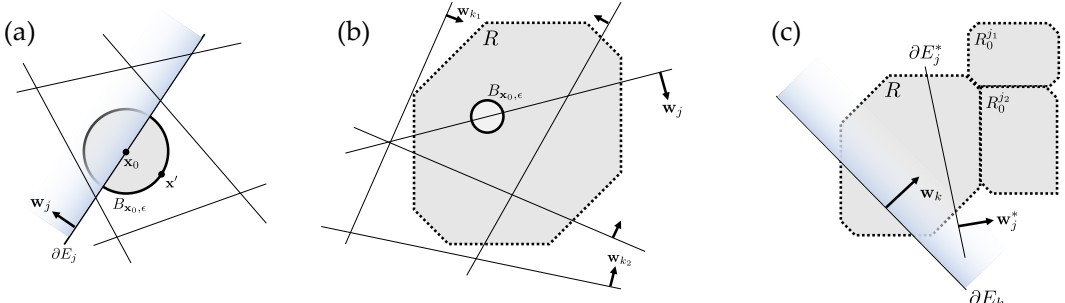

Figure 9: Proof illustration for (a) Lemma 2, (b) Lemma 3 and (c) Theorem 4.

## 8.3 COROLLARY 1

*Proof.* The base case is that $V_L(\mathbf{x}) = V_L^*(\mathbf{x}) = I_{C \times C}$, which is constant (and thus piece-wise constant) over the entire input space. If for layer $l$, $V_l(\mathbf{x})$ and $V_l^*(\mathbf{x})$ are piece-wise constant, then by Eqn. 4 (rewrite it here):

$$V_{l-1}(\mathbf{x}) = V_l(\mathbf{x})D_l(\mathbf{x})W_l^\mathsf{T}, \quad V_{l-1}^*(\mathbf{x}) = V_l^*(\mathbf{x})D_l^*(\mathbf{x})W_l^{*\mathsf{T}} \tag{21}$$

since $D_l(\mathbf{x})$ and $D_l^*(\mathbf{x})$ are piece-wise constant and $W_l^\mathsf{T}$ and $W_l^{*\mathsf{T}}$ are constant, we know that for layer $l-1$, $V_{l-1}(\mathbf{x})$ and $V_{l-1}^*(\mathbf{x})$ are piece-wise constant. Therefore, for all $l = 1, \ldots L$, $V_l(\mathbf{x})$ and $V_l^*(\mathbf{x})$ are piece-wise constant.

Therefore, $A_l(\mathbf{x})$ and $B_l(\mathbf{x})$ are piece-wise constant with respect to input $\mathbf{x}$. They separate the region $R_0$ into constant regions with boundary points in a zero-measured set. □

## 8.4 LEMMA 2

**Lemma 2.** *Consider $K$ ReLU activation functions $f_j(\mathbf{x}) = \sigma(\mathbf{w}_j^\mathsf{T}\mathbf{x})$ for $j = 1 \ldots K$. If $\mathbf{w}_j \neq 0$ and no two weights are co-linear, then $\sum_{j'} c_{j'} f_{j'}(\mathbf{x}) = 0$ for all $\mathbf{x} \in \mathbb{R}^{d+1}$ suggests that all $c_j = 0$.*

*Proof.* Suppose there exists some $c_j \neq 0$ so that $\sum_j c_j f_j(\mathbf{x}) = 0$ for all $\mathbf{x}$. Pick a point $\mathbf{x}_0 \in \partial E_j$ so that $\mathbf{w}_j^\mathsf{T}\mathbf{x}_0 = 0$ but all $\mathbf{w}_{j'}^\mathsf{T}\mathbf{x}_0 \neq 0$ for $j' \neq j$, which is possible due to the distinct weight conditions. Consider an $\epsilon$-ball $B_{\mathbf{x}_0,\epsilon} = \{\mathbf{x} : \|\mathbf{x} - \mathbf{x}_0\| \leq \epsilon\}$. We pick $\epsilon$ so that $\text{sign}(\mathbf{w}_{j'}^\mathsf{T}\mathbf{x})$ for all $j' \neq j$ remains the same within $B_{\mathbf{x}_0,\epsilon}$ (Fig. 9(a)). Denote $[j^+]$ as the indices of activated ReLU functions in $B_{\mathbf{x}_0,\epsilon}$ except $j$.

Then for all $\mathbf{x} \in B_{\mathbf{x}_0,\epsilon} \cap E_j$, we have:

$$h(\mathbf{x}) \equiv \sum_{j'} c_{j'} f_{j'}(\mathbf{x}) = c_j \mathbf{w}_j^\mathsf{T}\mathbf{x} + \sum_{j' \in [j^+]} c_{j'} \mathbf{w}_{j'}^\mathsf{T}\mathbf{x} = 0 \tag{22}$$

Since $B_{\mathbf{x}_0,\epsilon}$ is a $d$-dimensional object rather than a subspace, for $\mathbf{x}_0$ and $\mathbf{x}_0 + \epsilon\mathbf{e}_k \in B(\mathbf{x}_0, \epsilon)$, we have

$$h(\mathbf{x}_0 + \epsilon\mathbf{e}_k) - h(\mathbf{x}_0) = \epsilon(c_j w_{jk} + \sum_{j' \in [j^+]} c_{j'} w_{j'k}) = 0 \tag{23}$$

where $\mathbf{e}_k$ is axis-aligned unit vector ($1 \leq k \leq d$). This yields

$$c_j \tilde{\mathbf{w}}_j + \sum_{j' \in [j^+]} c_{j'} \tilde{\mathbf{w}}_{j'} = \mathbf{0}_d \tag{24}$$

Plug it back to Eqn. 22 yields

$$c_j b_j + \sum_{j' \in [j^+]} c_{j'} b_{j'} = 0 \tag{25}$$

where means that for the (augmented) $d + 1$ dimensional weight:

$$c_j \mathbf{w}_j + \sum_{j' \in [j^+]} c_{j'} \mathbf{w}_{j'} = \mathbf{0}_{d+1} \tag{26}$$

However, if we pick $\mathbf{x}' = \mathbf{x}_0 - \epsilon \frac{\tilde{\mathbf{w}}_j}{\|\tilde{\mathbf{w}}_j\|^2} \in B_{\mathbf{x}_0,\epsilon} \cap E_j^\complement$, then $f_j(\mathbf{x}') = 0$ but $\sum_{j' \in [j+]} f_j'(\mathbf{x}') = -c_j \mathbf{w}_j^\mathsf{T} \mathbf{x}' = \epsilon c_j$ and thus

$$\sum_{j'} c_{j'} f_{j'}(\mathbf{x}') = \epsilon c_j \neq 0 \tag{27}$$

which is a contradiction. $\qquad\square$

## 8.5 LEMMA 3

**Lemma 3** (Local ReLU Independence). *Let $R$ be an open set. Consider $K$ ReLU nodes $f_j(\mathbf{x}) = \sigma(\mathbf{w}_j^\mathsf{T}\mathbf{x})$, $j = 1, \ldots, K$. $\mathbf{w}_j \neq 0$, $\mathbf{w}_j \neq \gamma \mathbf{w}_{j'}$ for $j \neq j'$ with any $\gamma > 0$.*

*If there exists $c_1, \ldots, c_K, c_\bullet$ so that the following is true:*

$$\sum_j c_j f_j(\mathbf{x}) + c_\bullet \mathbf{w}_\bullet^\mathsf{T}\mathbf{x} = \mathbf{0}, \quad \forall \mathbf{x} \in R \tag{28}$$

*and for node $j$, $\partial E_j \cap R \neq \emptyset$, then $c_j = 0$.*

*Proof.* We can apply the same logic as Lemma 2 to the region $R$ (Fig. 9(b)). For any node $j$, since its boundary $\partial E_j$ is in $R$, we can find a similar $\mathbf{x}_0$ so that $\mathbf{x}_0 \in \partial E_j \cap R$ and $\mathbf{x}_0 \notin \partial E_{j'}$ for any $j' \neq j$. We construct $B_{\mathbf{x}_0,\epsilon}$. Since $R$ is an open set, we can always find $\epsilon > 0$ so that $B_{\mathbf{x}_0,\epsilon} \subseteq R$ and no other boundary is in this $\epsilon$-ball. Following similar logic of Lemma 2, $c_j = 0$. $\qquad\square$

## 8.6 LEMMA 4

**Lemma 4** (Relation between Hyperplanes). *Let $\mathbf{w}_j$ and $\mathbf{w}_{j'}$ two distinct hyperplanes with $\|\tilde{\mathbf{w}}_j\| = \|\tilde{\mathbf{w}}_{j'}\| = 1$. Denote $\theta_{jj'}$ as the angle between the two vectors $\mathbf{w}_j$ and $\mathbf{w}_{j'}$. Then there exists $\tilde{\mathbf{u}}_{j'} \perp \tilde{\mathbf{w}}_j$ and $\mathbf{w}_{j'}^\mathsf{T}\tilde{\mathbf{u}}_{j'} = \sin\theta_{jj'}$.*

*Proof.* Note that the projection of $\tilde{\mathbf{w}}_{j'}$ onto $\tilde{\mathbf{w}}_j$ is:

$$\tilde{\mathbf{u}}_{j'} = \frac{1}{\sin\theta_{jj'}} P_{\tilde{\mathbf{w}}_j}^\perp \tilde{\mathbf{w}}_{j'} \tag{29}$$

It is easy to verify that $\|\tilde{\mathbf{u}}_{j'}\| = 1$ and $\mathbf{w}_{j'}^\mathsf{T}\tilde{\mathbf{u}}_{j'} = \sin\theta_{jj'}$. $\qquad\square$

## 8.7 LEMMA 5

**Lemma 5** (Evidence of Data points on Misalignment). *Let $R \subset \mathbb{R}^d$ be an open set. Consider $K$ ReLU nodes $f_j(\mathbf{x}) = \sigma(\mathbf{w}_j^\mathsf{T}\mathbf{x})$, $j = 1, \ldots, K$. $\|\tilde{\mathbf{w}}_j\| = 1$, $\mathbf{w}_j$ are not co-linear. Then for a node $j$ with $\partial E_j \cap R \neq \emptyset$, and $\epsilon \leq \epsilon_0$, either of the conditions holds:*

*(1) There exists node $j' \neq j$ so that $\sin\theta_{jj'} \leq MK\epsilon^{1-\delta}/|c_j|$ and $|b_{j'} - b_j| \leq M_2\epsilon^{1-2\delta}/|c_j|$.*

*(2) There exists $\mathbf{x}_j \in \partial E_j \cap R$ so that for any $j' \neq j$, $|\mathbf{w}_{j'}^\mathsf{T}\mathbf{x}_j| > 5\epsilon/|c_j|$.*

*where $\theta_{jj'}$ is the angle between $\tilde{\mathbf{w}}_j$ and $\tilde{\mathbf{w}}_{j'}$, $\delta > 0$, $r$ is the radius of a $d-1$ dimensional ball contained in $\partial E_j \cap R$, $M = \frac{10\epsilon_0^\delta}{r}\sqrt{\frac{d}{2\pi}}$, $M_0 = \max_{\mathbf{x} \in \partial E_j \cap R} \|\mathbf{x}\|$ and $M_2 = 2M_0MK\epsilon_0^\delta + 5\epsilon_0^{2\delta}$.*

*Proof.* Define $q_j = 5\epsilon/|c_j|$. For each $j' \neq j$, define $I_{j'} = \{\mathbf{x} : |\mathbf{w}_{j'}^\mathsf{T}\mathbf{x}| \leq q_j, \mathbf{x} \in \partial E_j\}$. We prove by contradiction. Suppose for any $j' \neq j$, $\sin\theta_{jj'} > KM\epsilon^{1-\delta}/|c_j|$ or $|b_{j'} - b_j| > M_2\epsilon^{1-2\delta}/|c_j|$. Otherwise the theorem already holds.

**Case 1. When $\sin\theta_{jj'} > KM\epsilon^{1-\delta}/|c_j|$ holds.**

From Lemma 4, we know that for any $\mathbf{x} \in \partial E_j$, if $\mathbf{w}_{j'}^\mathsf{T}\mathbf{x} = -q_j$, with $a_{j'} \leq 2q_j|c_j|/MK\epsilon^{1-\delta} = 10\epsilon^\delta/MK$, we have $\mathbf{x}' = \mathbf{x} + a_{j'}\mathbf{u}_{j'} \in \partial E_j$ and $\mathbf{w}_{j'}^\mathsf{T}\mathbf{x}' = +q_j$.

Consider a $d-1$-dimensional sphere $B \subseteq \Omega_j$ and its intersection of $I_{j'} \cap B$ for $j' \neq j$. Suppose the sphere has radius $r$. For each $I_{j'} \cap B$, its $d-1$-dimensional volume is upper bounded by:

$$V(I_{j'} \cap B) \leq a_{j'} V_{d-2}(r) \leq \epsilon^\delta \frac{10}{MK} V_{d-2}(r) \tag{30}$$

where $V_{d-2}(r)$ is the $d-2$-dimensional volume of a sphere of radius $r$. Intuitively, the intersection between $\mathbf{w}_{j'}^\mathsf{T} \mathbf{x} = -q_j$ and $B$ is at most a $d-2$-dimensional sphere of radius $r$, and the "height" is at most $a_{j'}$.

**Case 2. When** $\sin \theta_{jj'} \leq KM\epsilon^{1-\delta}/|c_j|$ **but** $|b_{j'} - b_j| > M_2 \epsilon^{1-2\delta}/|c_j|$ **holds.**

In this case, we want to show that for any $\mathbf{x} \in \Omega_j$, $|\mathbf{w}_{j'}^\mathsf{T} \mathbf{x}| > q_j$ and thus $I_{j'} \cap B = \emptyset$. If this is not the case, then there exists $\mathbf{x} \in \Omega_j$ so that $|\mathbf{w}_{j'}^\mathsf{T} \mathbf{x}| \leq q_j$. Then since $\mathbf{x} \in \partial E_j$, we have:

$$|\mathbf{w}_{j'}^\mathsf{T} \mathbf{x}| = |(\mathbf{w}_{j'} - \mathbf{w}_j)^\mathsf{T} \mathbf{x}| = |(\tilde{\mathbf{w}}_{j'} - \tilde{\mathbf{w}}_j)^\mathsf{T} \tilde{\mathbf{x}} + (b_{j'}' - b_j)| \leq q_j \tag{31}$$

Therefore, from Cauchy inequality and triangle inequality, we have:

$$\|\tilde{\mathbf{w}}_{j'} - \tilde{\mathbf{w}}_j\|\|\tilde{\mathbf{x}}\| \geq |(\tilde{\mathbf{w}}_{j'} - \tilde{\mathbf{w}}_j)^\mathsf{T} \tilde{\mathbf{x}}| \geq |b_{j'}' - b_j| - |\mathbf{w}_{j'}^\mathsf{T} \mathbf{x}| \tag{32}$$

From the condition, we have $\|\tilde{\mathbf{w}}_{j'} - \tilde{\mathbf{w}}_j\| = 2 \sin \frac{\theta_{jj'}}{2} \leq 2 \sin \theta_{jj'} \leq 2KM\epsilon^{1-\delta}/|c_j|$. Then

$$2M_0 MK\epsilon^{1-\delta}/|c_j| \geq |(\tilde{\mathbf{w}}_{j'} - \tilde{\mathbf{w}}_j)^\mathsf{T} \tilde{\mathbf{x}}| \geq |b_{j'} - b_j| - q_j > M_2 \epsilon^{1-2\delta}/|c_j| - 5\epsilon/|c_j| \tag{33}$$

which is equivalent to:

$$2M_0 MK\epsilon^\delta > M_2 - 5\epsilon^{2\delta} \tag{34}$$

which means that

$$M_2 < 2M_0 MK\epsilon^\delta + 5\epsilon^{2\delta} \leq 2M_0 MK\epsilon_0^\delta + 5\epsilon_0^{2\delta} \tag{35}$$

for $\epsilon \leq \epsilon_0$. This is a contradiction. Therefore, $I_{j'} \cap B = \emptyset$ and thus $V(I_{j'} \cap B) = 0$.

**Volume argument.** Therefore, from the definition of $M$, we have $V(B) = V_{d-1}(r) \geq r\sqrt{\frac{2\pi}{d}} V_{d-2}(r) = \frac{10}{M} \epsilon_0^\delta V_{d-2}(r)$, then for $\epsilon \leq \epsilon_0$, we have:

$$V(B) = \frac{10}{M} \epsilon_0^\delta V_{d-2}(r) > \sum_{j' \neq j, j' \text{ in case 1}} V(I_{j'} \cap B) \tag{36}$$

This means that there exists $\mathbf{x}_j \in B \subseteq \Omega_j$ so that $\mathbf{x}_j \notin I_{j'} \cap B$ for any $j' \neq j$ and $j'$ in case 1. That is,

$$|\mathbf{w}_{j'}^\mathsf{T} \mathbf{x}_j| > q_j \tag{37}$$

On the other hand, for $j'$ in case 2, the above condition holds for entire $\Omega_j$, and thus hold for the chosen $\mathbf{x}_j$. $\qquad \square$

## 8.8 LEMMA 6

**Lemma 6** (Local ReLU Independence, Noisy case). *Let $R$ be an open set. Consider $K$ ReLU nodes $f_j(\mathbf{x}) = \sigma(\mathbf{w}_j^\mathsf{T} \mathbf{x})$, $j = 1, \ldots, K$. $\|\tilde{\mathbf{w}}_j\| = 1$, $\mathbf{w}_j$ are not co-linear. If there exists $c_1, \ldots, c_K, c_\bullet$ and $\epsilon \leq \epsilon_0$ so that the following is true:*

$$\left| \sum_j c_j f_j(\mathbf{x}) + c_\bullet \mathbf{w}_\bullet^\mathsf{T} \mathbf{x} \right| \leq \epsilon, \quad \forall \mathbf{x} \in R \tag{38}$$

*and for a node $j$, $\partial E_j \cap R \neq \emptyset$. Then there exists node $j' \neq j$ so that $\sin \theta_{jj'} \leq MK\epsilon^{1-\delta}/|c_j|$ and $|b_{j'} - b_j| \leq M_2 \epsilon^{1-2\delta}/|c_j|$, where $r, \delta, M, M_2$ are defined in Lemma 5 but with $r' = r - 5\epsilon/|c_j|$.*

*Proof.* Let $q_j = 5\epsilon/|c_j|$ and $\Omega_j = \{\mathbf{x} : \mathbf{x} \in \partial E_j \cap R, \ B(\mathbf{x}, q_j) \subseteq R\}$. If situation (1) in Lemma 5 happens then the theorem holds. Otherwise, applying Lemma 5 with $R' = \{\mathbf{x} : \mathbf{x} \in R, \ B(\mathbf{x}, q_j) \subseteq R\}$ and there exists $\mathbf{x}_j \in \Omega_j$ so that

$$|\mathbf{w}_{j'}^\mathsf{T} \mathbf{x}_j| \geq q_j = 5\epsilon/|c_j| \tag{39}$$

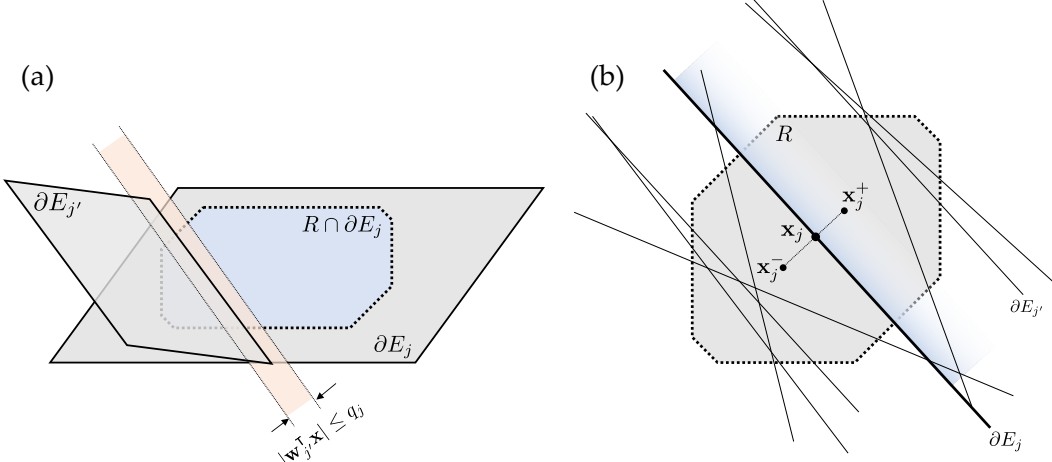

Figure 10: **(a)** Lemma 5. **(b)** Lemma 6.

Let two points $\mathbf{x}_j^{\pm} = \mathbf{x}_j \pm q_j \tilde{\mathbf{w}}_j \in R$. In the following we show that the three points $\mathbf{x}_j$ and $\mathbf{x}_j^{\pm}$ are on the same side of $\partial E_{j'}$ for any $j' \neq j$. This can be achieved by checking whether $(\mathbf{w}_{j'}^{\mathsf{T}}\mathbf{x}_j)(\mathbf{w}_{j'}^{\mathsf{T}}\mathbf{x}_j^{\pm}) \geq 0$ (Fig. 10):

$$
\begin{align}
(\mathbf{w}_{j'}^{\mathsf{T}}\mathbf{x}_j)(\mathbf{w}_{j'}^{\mathsf{T}}\mathbf{x}_j^{\pm}) &= (\mathbf{w}_{j'}^{\mathsf{T}}\mathbf{x}_j)\left[\mathbf{w}_{j'}^{\mathsf{T}}(\mathbf{x}_j \pm q_j\tilde{\mathbf{w}}_j)\right] \tag{40} \\
&= (\mathbf{w}_{j'}^{\mathsf{T}}\mathbf{x}_j)^2 \pm q_j(\mathbf{w}_{j'}^{\mathsf{T}}\mathbf{x}_j)\mathbf{w}_{j'}^{\mathsf{T}}\tilde{\mathbf{w}}_j \tag{41} \\
&= |\mathbf{w}_{j'}^{\mathsf{T}}\mathbf{x}_j|(|\mathbf{w}_{j'}^{\mathsf{T}}\mathbf{x}_j| \pm q_j\mathbf{w}_{j'}^{\mathsf{T}}\tilde{\mathbf{w}}_j) \tag{42}
\end{align}
$$

Since $|\mathbf{w}_{j'}^{\mathsf{T}}\tilde{\mathbf{w}}_j| \leq 1$, it is clear that $(\mathbf{w}_{j'}^{\mathsf{T}}\mathbf{x}_j)(\mathbf{w}_{j'}^{\mathsf{T}}\mathbf{x}_j^{\pm}) \geq 0$. Therefore the three points $\mathbf{x}_j$ and $\mathbf{x}_j^{\pm}$ are on the same side of $\partial E_{j'}$ for any $j' \neq j$.

Let $h(\mathbf{x}) = \sum_j c_j f_j(\mathbf{x}) + c_{\bullet}\mathbf{w}_{\bullet}^{\mathsf{T}}\mathbf{x}$, then $|h(\mathbf{x})| \leq \epsilon$ for $\mathbf{x} \in R$. Since $\mathbf{x}_j^{+} + \mathbf{x}_j^{-} = 2\mathbf{x}_j$, we know that all terms related to $\mathbf{w}_{\bullet}$ and $\mathbf{w}_{j'}$ with $j \neq j$ will cancel out (they are in the same side of the boundary $\partial E_{j'}$) and thus:

$$
4\epsilon \geq |h(\mathbf{x}_j^{+}) + h(\mathbf{x}_j^{-}) - 2h(\mathbf{x}_j)| = |c_j q_j \mathbf{w}_j^{\mathsf{T}}\mathbf{w}_j| = |c_j|q_j = 5\epsilon \tag{43}
$$

which is a contradiction. $\qquad\square$

## 8.9 THEOREM 2

*Proof.* In this situation, because $D_2(\mathbf{x}) = D_2^*(\mathbf{x}) = I$, according to Eqn. 4, $V_1(\mathbf{x}) = W_1^{\mathsf{T}}$ and $V_1^*(\mathbf{x}) = W_1^{*\mathsf{T}}$ are independent of input $\mathbf{x}$. Therefore, both $A_1$ and $B_1$ are independent of input $\mathbf{x}$.

From Assumption 1, since $\rho(\mathbf{x}) > 0$ in $R_0$, from Theorem. 1 we know that the SGD critical points gives $\mathbf{g}_1(\mathbf{x}) = D_1(\mathbf{x})\left[A_1\mathbf{f}_1^*(\mathbf{x}) - B_1\mathbf{f}_1(\mathbf{x})\right] = \mathbf{0}$. Picking node $k$, the following holds for every node $k$ and every $\mathbf{x} \in R_0 \cap E_k$:

$$
\boldsymbol{\alpha}_k^{\mathsf{T}}\mathbf{f}^*(\mathbf{x}) - \boldsymbol{\beta}_k^{\mathsf{T}}\mathbf{f}(\mathbf{x}) = \mathbf{0} \tag{44}
$$

Here $\boldsymbol{\alpha}_k^{\mathsf{T}}$ is the $k$-th row of $A_1$, $A_1 = [\boldsymbol{\alpha}_1, \ldots, \boldsymbol{\alpha}_{n_1}]^{\mathsf{T}}$ and similarly for $\boldsymbol{\beta}_k^{\mathsf{T}}$. Note here layer index $l = 1$ is omitted for brevity.

For teacher $j$, suppose it is observed by student $k$, i.e., $\partial E_j^* \cap E_k \neq \emptyset$. Given all teacher and student nodes, note that co-linearity is a equivalent relation, we could partition these nodes into disjoint groups. Suppose node $j$ is in group $s$. In Eqn. 44, if we combine all coefficients in group $s$ together into one term $c_s\mathbf{w}_j^*$ (with $\|\mathbf{w}_j^*\| = 1$), we have:

$$
c_s = \alpha_{kj} - \sum_{k' \in \text{co-linear}(j)} \|\mathbf{w}_{k'}\|\beta_{kk'} \tag{45}
$$

"At most" because from Assumption 1, all teacher weights are not co-linear. Note that co-linear$(j)$ might be an empty set.

By Assumption 1, $\partial E_j^* \cap R_0 \neq \emptyset$ and by observation property, $\partial E_j^* \cap E_k \neq \emptyset$, we know that for $R = R_0 \cap E_k$, $\partial E_j^* \cap R \neq \emptyset$. Applying Lemma 3, we know that $c_s = 0$. Since $\alpha_{kj} \neq 0$, we know co-linear$(j) \neq \emptyset$ and there exists at least one student $k'$ that is aligned with the teacher $j$. $\qquad \square$

## 8.10 THEOREM 3

*Proof.* We basically apply the same logic as in Theorem 2. Consider the colinear group co-linear$(k)$. If for all $k' \in$ co-linear$(k)$, $\beta_{k'k'} \equiv \|\mathbf{v}_{k'}\|^2 = 0$, then $\mathbf{v}_{k'} = \mathbf{0}$ and the proof is complete.

Otherwise, if there exists some student $k$ so that $\mathbf{v}_k \neq \mathbf{0}$. By the condition, it is observed by some student node $k_o$, then with the same logic we will have

$$\sum_{k' \in \text{co-linear}(k)} \beta_{k_o,k'} \|\mathbf{w}_{k'}\| = 0 \tag{46}$$

which is

$$\mathbf{v}_{k_o}^\mathsf{T} \sum_{k' \in \text{co-linear}(k)} \mathbf{v}_{k'} \|\mathbf{w}_{k'}\| = 0 \tag{47}$$

Since $k$ is observed by $C$ students $k_o^1, k_o^2, \ldots, k_o^J$, then we have:

$$\mathbf{v}_{k_o^j}^\mathsf{T} \sum_{k' \in \text{co-linear}(k)} \mathbf{v}_{k'} \|\mathbf{w}_{k'}\| = 0 \tag{48}$$

By the condition, all the $C$ vectors $\mathbf{v}_{k_o^j}^\mathsf{T} \in \mathbb{R}^C$ are linear independent, then we know that

$$\sum_{k' \in \text{co-linear}(k)} \mathbf{v}_{k'} \|\mathbf{w}_{k'}\| = \mathbf{0} \tag{49}$$

$\square$

## 8.11 COROLLARY 2

*Proof.* We can write the contribution of all student nodes which are not aligned with any teacher nodes as follows:

$$\sum_s \sum_{k \in \text{co-linear}(s)} \mathbf{v}_k f_k(\mathbf{x}) = \sum_s \sum_{k \in \text{co-linear}(s)} \mathbf{v}_k \|\mathbf{w}_k\| \sigma(\mathbf{w}_s'^\mathsf{T} \mathbf{x}) \tag{50}$$

$$= \sum_s \sigma(\mathbf{w}_s'^\mathsf{T} \mathbf{x}) \sum_{k \in \text{co-linear}(s)} \mathbf{v}_k \|\mathbf{w}_k\| \tag{51}$$

where $\mathbf{w}_s'$ is the unit vector that represents the common direction of the co-linear group $s$. From Theorem 3, for group $s$ that is not aligned with any teacher, $\sum_{k \in \text{co-linear}(s)} \mathbf{v}_k \|\mathbf{w}_k\| = \mathbf{0}$ and thus the net contribution is zero. $\qquad \square$

## 8.12 THEOREM 4

*Proof.* In multi-layer case, $A_l(\mathbf{x})$ and $B_l(\mathbf{x})$ are no longer constant over input $\mathbf{x}$. Fortunately, thanks to the recursive definition (Eqn. 4) which only contains input-independent terms (weights) and gating function, $A_l(\mathbf{x})$ and $B_l(\mathbf{x})$ are piece-wise constant function over the input $R_0$.

Note that $R_0$ can be partitioned into $\mathcal{R} = \{R_0^1, R_0^2, \ldots, R_0^J\}$ and a zero-measure set. Each of them is constant region for $A_l(\mathbf{x})$ and $B_l(\mathbf{x})$. Since $R_0^j$ is an intersection of finite open half-planes (from $k$'s parent nodes), $R_0^j$ is still an open set.

From the condition, there exists open set $R \in \mathcal{R}$ and a student observer node $k$ so that $\partial E_j^* \cap E_k \cap R \neq \emptyset$ ((Fig. 9(c)). Let $H_R$ and similarly $H_R^*$ be the student and teacher nodes whose boundary

intersects with $R$. Therefore $j \in H_R^*$. For other teacher/student nodes, they are linear functions within $R$ and thus can be combined together into $\mathbf{w}_\bullet^\intercal \mathbf{x}$. For all weights in $H_R$, $H_R^*$ and $\mathbf{w}_\bullet$, applying Lemma 3 on $R \cap E_k$, we know that the SGD critical point $\boldsymbol{\alpha}_{R,k}^\intercal \mathbf{f}_1^*(\mathbf{x}) - \boldsymbol{\beta}_{R,k}^\intercal \mathbf{f}_1(\mathbf{x}) = \mathbf{0}$ leads to alignment between $H_R$ and $H_R^*$. Let group $s$ be the one that contains all weights that are co-linear to teacher node $j$ (note that no other teacher nodes are involved), and $c_s$ its coefficient. Since $j \in H_R^*$, $c_s = 0$. Since $\alpha_{kj}(R) \neq 0$, there exists at least one student node $k'$ that is co-linear to teacher node $j$.

$\square$

## 8.13 THEOREM 5

*Proof.* We follow the logic of Theorem 4. Instead of applying Lemma 3, for gradient that is not zero but bounded within $\epsilon$, we pick the student observer $k$ and we have for $E_k \cap R$:

$$|\boldsymbol{\alpha}_k^\intercal \mathbf{f}^*(\mathbf{x}) - \boldsymbol{\beta}_k^\intercal \mathbf{f}(\mathbf{x})| \leq \epsilon, \tag{52}$$

we use Lemma 6 and know that there exists a node $k' \neq j$ so that $\sin \theta_{k'j} = \mathcal{O}\left(\epsilon^{1-\delta}/|c_j|\right)$ and $|b_{k'} - b_j^*| = \mathcal{O}\left(\epsilon^{1-2\delta}/|c_j|\right)$ for any $\delta > 0$. Under the observation of student $k$, the teacher $j$ has coefficient $c_j = \alpha_{kj}$. Since all teacher weights are distant to each other with positive constant $b_0 > 0$ and $\theta_0 > 0$, with sufficiently small $\epsilon_0$ and $\epsilon \leq \epsilon_0$, this node $k'$ has to be a student node and the bound follows. $\square$

## 8.14 THEOREM 6

*Proof.* From the expression we can see that it is positive homogeneous with respect to $\|\mathbf{w}_j^*\|$ and $\|\mathbf{w}_k\|$. So we can assume $\|\mathbf{w}_j^*\| = \|\mathbf{w}_k\| = 1$. Without loss of generality, we set up the coordinate system so that $\mathbf{w}_j^* = [1, 0]^\intercal$ and $\mathbf{w}_k = [\cos \theta, \sin \theta]^\intercal$. Then

$$\mathbb{E}_\mathbf{x}\left[\mathbf{f}_{l-1} z_k f_j^*\right] \;=\; \mathbb{E}_\mathbf{x}\left[\mathbf{f}_{l-1} z_k z_j^* \mathbf{f}_{l-1}^\intercal\right] \mathbf{w}_j^* = \sum_{\mathbf{f}_{l-1}^\intercal \mathbf{w}_j^* \geq 0,\, \mathbf{f}_{l-1}^\intercal \mathbf{w}_k \geq 0} \mathbf{f}_{l-1} \mathbf{f}_{l-1}^\intercal \mathbf{w}_j^* \tag{53}$$

$$= \int_0^{+\infty} r^2 p(r) \int_{-\frac{\pi}{2}+\theta}^{\frac{\pi}{2}} \begin{bmatrix} \cos \theta' \\ \sin \theta' \end{bmatrix} \cos \theta' p(\theta'|r) \mathrm{d}\theta' + \boldsymbol{\epsilon} \tag{54}$$

where $\boldsymbol{\epsilon}$ is the term reflecting the asymmetry of the data distribution $p(\mathbf{f}_{l-1})$ with respect to the plane spanned by the vectors $\mathbf{w}_k$ and $\mathbf{w}_j^*$.

If the data distribution $p(\mathbf{f}_{l-1})$ is scale invariant (rescaling the data point won't change the angular distribution), then $p(\theta'|r) = p(\theta')$ and we only need to check the angular integral:

$$\mathbf{I}(\theta) = \int_{-\frac{\pi}{2}+\theta}^{\frac{\pi}{2}} \begin{bmatrix} \cos \theta' \\ \sin \theta' \end{bmatrix} \cos \theta' p(\theta') \mathrm{d}\theta' \tag{55}$$

Note that $\cos^2 \theta = \frac{1}{2}(1 + \cos 2\theta)$ and $\sin \theta \cos \theta = \frac{1}{2}\sin 2\theta$, so we have:

$$2\mathbf{I}(\theta) \;=\; \left(\int_{-\frac{\pi}{2}+\theta}^{\frac{\pi}{2}} p(\theta')\mathrm{d}\theta'\right) \mathbf{w}_j^* + \int_{-\frac{\pi}{2}+\theta}^{\frac{\pi}{2}} \begin{bmatrix} \cos 2\theta' \\ \sin 2\theta' \end{bmatrix} p(\theta')\mathrm{d}\theta' \tag{56}$$

$$= \left(\int_{-\frac{\pi}{2}+\theta}^{\frac{\pi}{2}} p(\theta')\mathrm{d}\theta'\right) \mathbf{w}_j^* + \frac{1}{2}\int_{2\theta}^{2\pi} \begin{bmatrix} \cos \theta'' \\ \sin \theta'' \end{bmatrix} p\left(\frac{\theta''}{2} - \frac{\pi}{2}\right) \mathrm{d}\theta'' \tag{57}$$

$$= I_1(\theta)\mathbf{w}_j^* + \frac{1}{2}\mathbf{I}_0 - \frac{1}{2}\mathbf{I}_2(\theta) \tag{58}$$

where $\theta'' = 2\theta' + \pi$ and

$$I_1(\theta) \ = \ \int_{-\frac{\pi}{2}+\theta}^{\frac{\pi}{2}} p(\theta')\mathrm{d}\theta' \tag{59}$$

$$\mathbf{I}_0 \ = \ \int_0^{2\pi} \begin{bmatrix} \cos\theta'' \\ \sin\theta'' \end{bmatrix} p\left(\frac{\theta''}{2} - \frac{\pi}{2}\right) \mathrm{d}\theta'' \tag{60}$$

$$\mathbf{I}_2(\theta) \ = \ \int_0^{2\theta} \begin{bmatrix} \cos\theta'' \\ \sin\theta'' \end{bmatrix} p\left(\frac{\theta''}{2} - \frac{\pi}{2}\right) \mathrm{d}\theta''$$

$$= \ \left\{ \int_0^\theta \left[ p\left(\frac{\theta'}{2} - \frac{\pi}{2}\right) + p\left(\theta - \frac{\theta'}{2} - \frac{\pi}{2}\right) \right] \cos\theta' \mathrm{d}\theta' \right\} \mathbf{w}_k$$

$$+ \ \left\{ \int_0^\theta \left[ p\left(\frac{\theta'}{2} - \frac{\pi}{2}\right) - p\left(\theta - \frac{\theta'}{2} - \frac{\pi}{2}\right) \right] \sin\theta' \mathrm{d}\theta' \right\} \mathbf{w}_k^\perp \tag{61}$$

where $\mathbf{w}_k^\perp$ is the unit vector that is perpendicular to $\mathbf{w}_k$ but still in the plane spanned by $\mathbf{w}_k$ and $\mathbf{w}_j^*$. Note $\mathbf{I}_0$ is the fixed integral of unit vectors weighted by angular distribution of input data on activated half-plane $E_j^*$ of teacher node $j$.

If $p(\mathbf{f}_{l-1})$ is rotational symmetric, then $\epsilon = 0$, $p(\theta') = \frac{1}{2\pi}$, then we can compute these terms analytically: $\mathbf{I}_0 = \mathbf{0}$, $I_1(\theta) = \frac{1}{2\pi}(\pi - \theta)$ and $\mathbf{I}_2(\theta) = \frac{1}{\pi}\sin\theta\,\mathbf{w}_k$. $\qquad\square$

## 8.15  THEOREM 7

*Proof.* Note that we have:

$$\frac{\mathrm{d}}{\mathrm{d}t}\|\mathbf{w}_k\| = \frac{\mathrm{d}}{\mathrm{d}t}\sqrt{\|\mathbf{w}_k\|^2} = \frac{2\mathbf{w}_k^\mathsf{T}\dot{\mathbf{w}}_k}{2\|\mathbf{w}_k\|} = \frac{1}{\|\mathbf{w}_k\|}\mathbf{w}_k^\mathsf{T}\dot{\mathbf{w}}_k = \mathbf{w}_k^\mathsf{T}\mathbf{r}_k \tag{62}$$

Therefore, we have

$$\frac{\mathrm{d}}{\mathrm{d}t}\ln\|\mathbf{w}_k\| = \bar{\mathbf{w}}_k^\mathsf{T}\mathbf{r}_k \tag{63}$$

and

$$\frac{\mathrm{d}}{\mathrm{d}t}\left(\ln\frac{\|\mathbf{w}_k\|}{\|\mathbf{w}_{k'}\|}\right) = \frac{\mathrm{d}}{\mathrm{d}t}(\ln\|\mathbf{w}_k\| - \ln\|\mathbf{w}_{k'}\|) = \bar{\mathbf{w}}_k^\mathsf{T}\mathbf{r}_k - \bar{\mathbf{w}}_{k'}^\mathsf{T}\mathbf{r}_{k'} \tag{64}$$

Note that we have:

$$\frac{\mathrm{d}}{\mathrm{d}t}\bar{\mathbf{w}}_k = \frac{\mathrm{d}}{\mathrm{d}t}\left(\frac{\mathbf{w}_k}{\|\mathbf{w}_k\|}\right) = \mathbf{r}_k - \mathbf{w}_k\frac{\mathbf{w}_k^\mathsf{T}\mathbf{r}_k}{\|\mathbf{w}_k\|^2} = (I - \bar{\mathbf{w}}_k\bar{\mathbf{w}}_k^\mathsf{T})\mathbf{r}_k = P_{\mathbf{w}_k}^\perp\mathbf{r}_k \tag{65}$$

Let $h_k = \bar{\mathbf{w}}_k^\mathsf{T}\mathbf{r}_k$. We assume all $h_k > 0$ (positive correlation), then we have:

$$\frac{\mathrm{d}}{\mathrm{d}t}h_k = \mathbf{r}_k^\mathsf{T}P_{\mathbf{w}_k}^\perp\mathbf{r}_k + \bar{\mathbf{w}}_k^\mathsf{T}\dot{\mathbf{r}}_k = \|\mathbf{r}_k\|^2 - h_k^2 + \bar{\mathbf{w}}_k^\mathsf{T}\dot{\mathbf{r}}_k \tag{66}$$

If $\mathbf{r}_k = \mathbf{r} = \mathbf{w}^* - \sum_k a_k\mathbf{w}_k$, then we have:

$$\frac{\mathrm{d}}{\mathrm{d}t}h_k = \|\mathbf{r}\|^2 - h_k^2 - Sh_k \tag{67}$$

where $S = \left(\sum_k a_k\|\mathbf{w}_k\|\right) > 0$ is independent of $k$. So

$$\frac{\mathrm{d}}{\mathrm{d}t}(h_k - h_{k'}) = (h_{k'}^2 - h_k^2) + S(h_{k'} - h_k) = (h_{k'} - h_k)(h_{k'} + h_k + S) \tag{68}$$

if $h_k - h_{k'} > 0$, then $\frac{\mathrm{d}}{\mathrm{d}t}(h_k - h_{k'}) < 0$ and vice versa. This means that Eqn. 64 is zero when the system enters the stable region. On the other hand, if $\|\mathbf{r}_k\|^2 = \|\mathbf{r}_{k'}\|^2 + \epsilon$ (e.g., $\mathbf{r}_k$ has stronger teacher component), then we have:

$$\frac{\mathrm{d}}{\mathrm{d}t}(h_k - h_{k'}) = (h_{k'} - h_k)(h_{k'} + h_k + S) + \epsilon \tag{69}$$

which is only zero when $h_k > h_{k'}$. This yields exponential growth of $\|\mathbf{w}_k\|$ compared to $\|\mathbf{w}_{k'}\|$. $\quad\square$

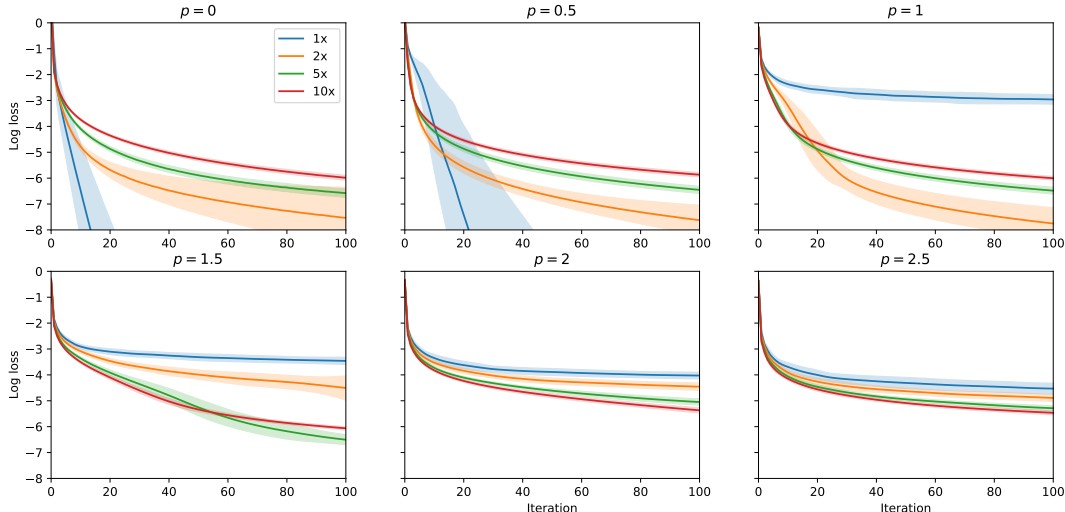

Figure 11: Evaluation loss convergence curve.

## 8.16 Details in Teacher/Student Construction and Training

We construct teacher networks in the following manner. For two-layered network, the output dimension $C = 50$ and input dimension $d = m_0 = n_0 = 100$. For multi-layered network, we use 50-75-100-125 (i.e, $m_1 = 50, m_2 = 75, m_3 = 100, m_4 = 125, L = 5, d = m_0 = n_0 = 100$ and $C = m_5 = n_5 = 50$). The teacher network is constructed to satisfy Assumption 1: at each layer, teacher filters are distinct from each other and their bias is set so that $\sim 50\%$ of the input data activate the nodes. This makes their boundary (maximally) visible in the dataset.

To train the model, we use vanilla SGD with learning rate $0.01$ and batchsize $16$.

## 8.17 Additional Figures

Fig. 11 shows how the loss changes over iterations. With high teacher polarity (Eqn. 9), it becomes harder to learn the weak teacher nodes and over-realization helps in getting low evaluation loss (in particular for $p = 2.5$).

Besides Gaussian distribution we also test on uniform distribution $\mathbf{x} \sim U[-15, 15]$. For training, we sample $100k$ data points in each epoch. Fig. 12 shows that the results on 4 layer ReLU network (50-75-100-125) are similar. Note that in multi-layer setting, Theorem 3 might not hold since it is for 2-layer so there could be un-specialized student nodes with large $\beta_{kk}(\mathbf{x})$.

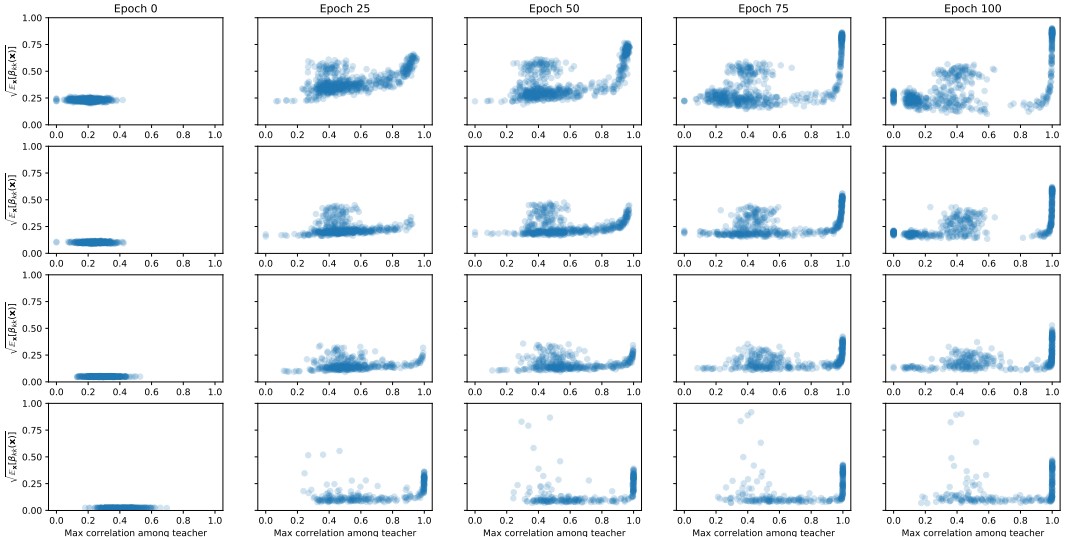

Figure 12: Strength of student specialization for 4 layer network (50-75-100-125) when each entry of the input dimension is uniform distributed in $U[-15, 15]$. For all teacher nodes, the normalized correlations are all close to $1.0$ ($\rho_{\mathrm{mean}} \geq 0.998$ at all layers).

