# OpenReview forum: "Toward Understanding Generalization of Over-parameterized Deep ReLU network trained with SGD in Student-teacher Setting"
_ICLR.cc/2020/Conference — Reject_

### Official Review · AnonReviewer2 · 2019-10-22
**Official Blind Review #2**

**Rating:** 3

**Review:**

The paper is well written, but I am not entirely sure of the interest of the results.
I might accept, but would not be too disappointed if it didn't pass.

A first comment is that the  alignment between student and teacher nodes is a very old problem, discussed at length in, for instance Saad&Solla, under the name "specialisation". Since the phenomenon is known, and already has a name, it should at least be also refereed to as such.

The result on the overlap and the "specialisation" of the teacher to the student presented in the paper is rigorous (though I did not completely checked the proof), and seems general enough, but it seems a bit trivial: of course if I have no or little error on all my data-points, I have overlap with the teacher, and since I'm over-parameterised and it's a ReLU network, then the alignment will be many-to-one.  More interesting would be to study alignment in the deeper case, but the authors prove it only for the lowest layer of a deep network.

The paper is mainly mathematical, and they are number of things I would find more interesting than the proof (though of course, this is a personal bias):
- plot the overlaps layer-wise (i.e. student layer 1 vs teacher layer 1, student layer 2 vs teacher layer 2, etc.) What do they look like? That's something I would actually quite like to know!
- the result on larger nodes being learnt first is known for online learning already in the 1990s (This is a celebrated results of Saad&Solla, though not an entirely rigorous one, and only for model data), so here the contribution is to show this for ReLU networks in particular.
- Since ReLU networks are somewhat linear, it would be interesting to compare the results on the dynamics to plain linear networks, as in Saxe et al (e.g. https://arxiv.org/abs/1710.03667 ). Discuss similarities / differences?
- The absence of "specialisation" in linear model is also a well known feature, see for instance https://arxiv.org/abs/1312.6120 https://arxiv.org/abs/1710.03667.

Finally, I am a bit confused by the experiments : I did not understand which experiment is done for which data in fig. 5.6.7 (8 is for CIFAR of course) and 11.



**Experience Assessment:**

I have read many papers in this area.

**Review Assessment: Checking Correctness Of Derivations And Theory:**

I did not assess the derivations or theory.

**Review Assessment: Checking Correctness Of Experiments:**

I assessed the sensibility of the experiments.

**Review Assessment: Thoroughness In Paper Reading:**

I read the paper at least twice and used my best judgement in assessing the paper.

---

> ### Author Response · Authors · 2019-11-06
> **Thanks for your comments.**
>
> Thanks the reviewer for the comments.
>
> In the next revision we will definitely use the word "specialization" and reference relevant papers. We apologize our reinvention of terms and will correct them.
>
> Although the intuition from the reviewer is roughly correct, Theorem 2 and Theorem 5 are not trivial to see since both teacher and student are deep ReLU networks and there could be complicated interactions between their nodes. For example:
>
> 1. Little error at every data point doesn't mean that each student could align with the teacher automatically (maybe two students can jointly explain one teacher with small errors?).
>
> 2. Teacher boundary overlapping with the dataset doesn't mean the teacher can be reconstructed automatically, which is the reason why we introduce the notation of "observer".
>
> 3. Also if there is no piecewise constancy of A and B matrix, then it is not clear that Theorem 5 would hold for the lowest layer in the multilayer setting.
>
> 4. The fact that there exist unaligned student nodes is not obvious from the intuition but only appear after mathematical analysis.
>
> Overall, there is a gap between intuition and rigorous theorems and our paper tries to fill this gap, which is the contribution.
>
> Other issues:
>
> 1. plot the overlaps layer-wise
> Fig. 5 exactly shows that if you train a 4 hidden layer student network from a 4 hidden layer teacher, how would the student nodes specialize w.r.t the teacher. From the figure the specialization still happens and the bottom layer does that first.
>
> 2. larger nodes (or strong teacher) learned first
> Thanks for the reference, we will cite these works and compare with them. We apologize for missing these references.
>
> 3. Compare the dynamics to plain linear networks.
> We will provide a detailed comparison in the next revision. In general, the dynamics can be very different. Roughly speaking, without ReLU, we won't be able to see Theorem 6 holds (which encourage student nodes who are close to some teacher node to be closer in the training), and some interesting behaviors (e.g., slow convergence to weak teacher node) probably won't happen.
>
> 4. The absence of "specialisation" in linear model is also a well known feature
> We briefly mentioned that part but never treat it as our major contributions. We will reference the papers you have mentioned.
>
> 5. Experiments.
> For Fig. 5, 6 and 7, we randomly generate zero-mean Gaussian distributed data as input of both teacher and student. Note that our theoretical analysis doesn't assume the distribution. To show our analysis is agonistic to different distributions, in the next revision we will show the results with other distributions.

---

### Official Review · AnonReviewer1 · 2019-10-22
**Official Blind Review #1**

**Rating:** 3

**Review:**

The paper studies the role of over-parametrization in the student-teacher multilayer ReLU networks. It presents a theoretical part about properties of SGD critical points for the teacher-student setting. And a heuristic and empirical part on dynamics of the SDG algorithm as a function of properties of the teacher networks. Overall, given previous literature, I do not find the presented results novel nor fundamentally very interesting and some parts are hard to understand due to missing details. I tend to vote for rejection at this point. More detailed questions, comments follow.


In related works:

** Paragraph on "Teacher-student/realizable setting": The recent line of works is interesting, but the authors should be clearer about this being a very classical setting dating back several decades. The first paper I know where the teacher student setting appeared is by Garder, Derrida'83 (model B, https://iopscience.iop.org/article/10.1088/0305-4470/22/12/004/pdf). In the classical textbook on neural networks Engel, Andreas, and Christian Van den Broeck. Statistical mechanics of learning. Cambridge University Press, 2001, there is a very detailed account of many results on the setting from 80s and 90s.

** "A line of works (Saad & Solla, 1996; 1995; Goldt et al., 2019; Freeman & Saad, 1997; Mace & Coolen, 1998) studied the dynamics from a statistical mechanics point of view, focusing on local analysis near to some critical points." and "(Goldt et al., 2019) assumes Gaussian input and symmetric parameterization to analyze local structure around critical points," The statements that these works focus on local analysis is not correct. While some formal analysis in these works required an infinitesimally informed start toward the teacher the experiments (in particular all those in Goldt et al., 2019) are run from random initialization and these works show empirically that randomly initialized training converges exactly to the fixed points described in the analysis.

** "Local minima is Global" paragraph: This paragraph seems to neglect the empirically observed fact (e.g. https://arxiv.org/pdf/1906.02613.pdf) that there can be global minima that generalize bad. Hence being global does not ensure good generalization.


Body of the paper:

** The authors cite: "Previous works (Ge et al., 2017; Livni et al., 2014) show that empirically SGD does not recover the parameters of a teacher network up to permutation." but they fail to mention that separate line of work, e.g.  (Saad & Solla, 1996; 1995; Goldt et al., 2019) observed empirically the opposite.The different exiting works have to be reconciles and understood and that may be beyond the scope of the present work. But presenting only one side of the results is not helping.

** The part on the dynamics with strong and weak directions reminds me on the results on so called "INCREMENTAL LEARNING" e.g. in the work:
Andrew M Saxe, James L McClelland, and Surya Ganguli. Exact solutions to the nonlinear dynamics of learning in deep linear neural networks. arXiv preprint arXiv:1312.6120, 2013.
also later: https://arxiv.org/pdf/1809.10374.pdf and others.
It would be useful to understand what is the relation in more detail and comment on it.

** The experimental part of the paper has numerous flaws that make it hard to be understood. For instance the authors do not specify the distribution of the input data. Some experiments are run with CIFAR and others with "random" data, but random in which sense? While generalization is the main focus of the paper the experimental results focus on the alignments of the teacher and students without really being clear how specifically the speed or the generalization error improves when neural networks are overparametrised. I found this information only in Fig. 8 for the test error. In Fig. 11 I do not know what are the different panels. What is the parameter p? So I do not know what to conclude from this figure .... in the first pannel the non-overparametrized loss (blue) decreases fastest. In the last pannel all curves are comparable. But this would suggest that over-parametrizatoin is not really helping which seems to go agains the rest of the conclusion in the paper.

** A side remark: I note that the paper is on 10 pages and hence according to the paper call higher standards should be applied in the review process.





**Experience Assessment:**

I have published one or two papers in this area.

**Review Assessment: Checking Correctness Of Derivations And Theory:**

I assessed the sensibility of the derivations and theory.

**Review Assessment: Checking Correctness Of Experiments:**

I assessed the sensibility of the experiments.

**Review Assessment: Thoroughness In Paper Reading:**

I read the paper at least twice and used my best judgement in assessing the paper.

---

> ### Author Response · Authors · 2019-11-06
> **Thank you for your comments.**
>
> We really appreciate the reviewer for giving us a lot of related work in 80s and 90s. We are fully aware of these works and have cited some of them in the paper. In the next revision, we will have a dedicated paragraph to talk about this classic setting decades ago.
>
> In summary, the papers from 80s and 90s are all very interesting. However, they mainly address Gaussian input, step function/Gaussian erf as nonlinearity and/or a single trainable layer. Both the book "Statistical mechanics of learning" and Saad & Solla (e.g., On-line learning in soft committee machines, Phys Review, 1995) discuss situation when the input dimension goes to infinite (i.e., the thermodynamics limits). They also assume Gaussian erf as the non-linearity and only deal with two-layer networks.  In contrast, we focus on student-teacher setting in deep ReLU networks, showing rigorous theorems for student specialization in the lowest layer, with finite student width and finite input dimension.
>
> The link provided by the reviewer is a paper in 1989 rather than 1983.
>
> We acknowledge that (Goldt et al., 2019) empirically shows that random initialization yields specialization, which is consistent with our experiments. In the next revision, we will make the description more precise.
>
> Body of the paper:
>
> 1. We thank the reviewer for the additional references. These references are actually strengthening our paper since our main theorems (Theorem 2 and Theorem 5) show the alignment (or specialization) actually happens and in that paragraph we try to explain why (Ge et al., 2017; Livni et al., 2014) give negative results, possibly due to unaligned student nodes, etc.
>
> 2. Thanks the reviewer for bringing about these references. Both papers are dealing with deep linear networks, while our work is for deep ReLU networks. We will make clear connections in the next revision. As pointed out by Review 2, in linear model the alignment (or specialization) is absent, which is a big difference.
>
> 3. We sample zero-mean Gaussian distribution as the random input data, but our theoretical analysis doesn't assume the distribution. In Fig. 11, $p$ is defined in Section 6 (just before Section 6.1) as the polarity between strong and weak teacher node. When $p$ is small, all teacher nodes are similar and we don't need over-parameterization to yield better performance; when $p$ becomes large, there is a big difference between strong and weak teacher nodes and Fig. 11 shows the different in the evaluation error and overparameterization does a better job in the generalization (last figure in Fig. 11, red and green curves). Fig. 8 shows that more teacher nodes are covered by the student with large $p$. We will make the section more clear in the next revision.

---

> > ### Comment · AnonReviewer1 · 2019-11-15
> > **Acknowledgment of the revision**
> >
> > Dear Authors,
> >
> > I thank you for a detailed answer and revision you posted today. I am downloading it and will read it in detail, Unfortunately I will not be able to reach a conclusion today for other duties. I assure you that I will take the revision in detail consideration in the next stage of the discussion.

---

### Official Review · AnonReviewer3 · 2019-10-23
**Official Blind Review #3**

**Rating:** 3

**Review:**

This paper studies the learning of over-parameterized neural networks in the student-teacher setting. More specifically, this paper assumes that there is a fixed teacher network providing the output for student network to learn, where the student network is typically over-parameterized (i.e., wider than teacher network).

This paper first investigates the properties of critical points of student networks in the ideal case, i.e., assuming we have infinite number of training examples. Then the results have been generalized to a practical case (the gradient is smaller than some small quantity). Moreover, this paper further studies the training dynamics via gradient flow, and proves some convergence results of GD.

Overall, this paper is somewhat difficult to follow and understand. The notation system is kind of complicated and some assumptions seem to be unrealistic.  Detailed comments are as follows:

It is a little bit difficult to get insightful understandings towards the critical points of deep neural networks from the theorems provided in this paper. I would like to see clearer properties of the critical points learned by student network rather than some intermediate results.

The title is not consistent with the content of the paper. From the title of this paper looks like a characterization on the student network trained by SGD. However, throughout the paper, the authors somehow investigate the critical points under a stronger condition, i.e., all stochastic gradient is zero, rather than the widely used one, the expectation of stochastic gradient is zero. I don’t think the critical points considered in this paper can be guaranteed to be found by SGD. Besides, when analyzing the training dynamics, as provided in Section 5, the authors resort to gradient descent, because in (5) the dynamics of $W_k$ rely on the expectation of stochastic gradients.

Many statements should be elaborated in detail. For example, in the paragraph before Corollary 1, why $R_l$ is a convex polytope? In Theorem 2, what’s $\alpha_{kj}$? What’s the meaning of alignment? In the paragraph after Theorem 4, why Theorem 4 suggests a picture of bottom-up training? I believe the authors should provide a more detailed explanation.

This paper studies the over-parameterized student network, is there any condition on its width?

In Theorem 5, the assumption $\|g_1\|_\infty<\epsilon$ seems rather unrealistic, typically this bound can only hold in expectation or with high probability. Besides, why there is no condition on the sample size n in Theorem 5? It looks like Theorem 5 aims to tackle the case of finite number of training samples.

-----------------------------------
Thanks for your response and revision.  The current title is clearer and the definition of SGD critical points is more accurate. The observations regarding the alignment between teacher and student networks are indeed interesting. However, I still feel that this result is somehow difficult to parse, as I am not clear why this can be interpreted as the learning of the teacher network. Therefore I would like to keep my score.



**Experience Assessment:**

I have published one or two papers in this area.

**Review Assessment: Checking Correctness Of Derivations And Theory:**

I assessed the sensibility of the derivations and theory.

**Review Assessment: Checking Correctness Of Experiments:**

I assessed the sensibility of the experiments.

**Review Assessment: Thoroughness In Paper Reading:**

I read the paper at least twice and used my best judgement in assessing the paper.

---

> ### Author Response · Authors · 2019-11-05
> **Thanks for your comments.**
>
> We thank the reviewer for the comments.
>
> Note that we have clearly stated the difference between SGD and GD from Eqn. 1 and a sentence below "In SGD, the expectation is taken over a batch. In GD, it is over the entire dataset". So the same expectation symbol was overloaded for both GD and SGD and throughout the paper, we use SGD critical point conditions. Throughout the paper we follow this convention (same in the critical point analysis and training dynamics). The training dynamics also uses the same SGD expectation and there is no inconsistency.
>
> The per sample condition $g_i = 0$ is widely used in interpolation settings (check papers https://arxiv.org/abs/1712.06559, https://arxiv.org/abs/1810.13395 and https://arxiv.org/abs/1811.02564), and is not an exotic condition. When the SGD Batch gradient is zero, or expected stochastic gradient is zero (i.e., $E_{batch}[g] = 0$ or $\frac{1}{B}\sum_i g_i = 0$), we have already shown (in Definition 1 and Theorem 1 in Section 4.1) that it will lead to per sample condition $g_i = 0$. Similar logic can be applied to yield small $\|g_1\|$ when the expected stochastic gradient is small (up to a constant). On the other hand, previous works have already shown GD will get trapped into local minima even with Gaussian inputs.
>
> Theorem 2 and Theorem 5 show clear properties of trained student networks (Reviewer 2 also thinks it is general) and are served as one step towards a better understanding of networks.
>
> Different from many recent papers (e.g., NTK), our paper doesn't assume any condition on the width of the network to be trained. As long as the student width is larger than the teacher, Theorem 2 and Theorem 5 can be applied. Our work suggests that finite width is no longer an approximate of the infinite width case, but has its own interesting properties. We think this is one of the interesting part of the paper.
>
> Theorem 2 and Theorem 5 only holds for infinite number of samples ("a region of training set"). This is explicitly stated in the definition of $R_0$ (just above Corollary 1 in page 4), and in the last paragraph of page 6. We will leave finite sample analysis as the future work.

---

> > ### Author Response · Authors · 2019-11-05
> > **More elaboration in details.**
> >
> > We would love to elaborate more on these points but refrain from doing so due to 10 page limitation.
> >
> > 1. $R_l$ is not necessarily a convex polytope, but can be an arbitrary shape. We are very sorry for the confusion and will correct it in the next revision.   So the Corollary 1 now reads:
> >
> > "
> > $R_0$ can be decomposed into a finite (but potentially exponential) set of regions $\mathcal{R}_{l−1}$ = $\{R^1_{l-1}, R^2_{l-1}, \ldots, R^J_{l-1}\}$ plus a zero-measure set, so that $A_l(\mathbf{x})$ and $B_l(\mathbf{x})$ are constant within each region $R^j_{l-1}$ with respect to $\mathbf{x}$.
> > "
> >
> > The remaining part of the paper remains intact.
> >
> > 2. $\alpha_{kj}$ is the $j$-th entry of the vector $\alpha_k$, which is the $(k, j)$ element of matrix $A_1(\mathbf{x})$. We will make it more clear in the next revision.
> >
> > 3. Theorem 4 shows that if the gradient is zero (or small) at the lowest layer, the teacher and student could still align even if the top layer hasn't aligned yet. This suggests alignment could happen in bottom-up manner. Here we just show the intuition without a formal theorem (and build such theorems would deserve separate papers).

---

### Author Response · Authors · 2019-10-28
**Minor revision**

Note that throughout the paper, we actually don't need the "convex polytope" condition (as mentioned in the current version of Corollary 1) and the training region R_0 can be any shape (even if R_0 contains disconnected components). Sorry for the confusion and we will update the paper in the next revision.

---

### Public Comment · ~Yang_Yuan2 · 2019-11-01
**Very interesting result**

Thanks for the interesting result!
I feel the assumption that \partial Ej* cap Ek cap R is non-empty is a little bit strong (maybe still correct if the student network is huge). But the overall observations and claims are refreshing :)

---

> ### Author Response · Authors · 2019-11-04
> **Thanks for your interest!**
>
> Hi Yang,
>
> Thanks for your attention and glad that you like this paper. The condition $\partial E_j^* \cap E_k \cap R$ is not that strong considering that it only needs to be satisfied for one student node $k$ out of many student nodes in the over-parameterized student network, and one region $R$ (in multi-layer case) out of many regions intersecting with the teacher boundary $\partial E_j^*$, after training is complete.
>
> But we agree with you that it would be better if we could characterize when such condition happens. For now, what we can say is that over-parameterization could help since this is an existence argument. We leave a more formal characterization to the future work.

---

### Author Response · Authors · 2019-11-15
**Paper Revision**

Dear reviewers,

We really appreciate your insightful comments. Thanks for your work!

We have made a major revision of the paper and addressed the questions raised by the reviewers.

1. The title/abstract/introduction are updated to better position this paper among previous works. In particular, we propose a rigorous analysis for student specialization for deep ReLU networks in finite width and finite dimension setting, and concludes that when reaching SGD stationary point there exists student specialization in the lowest layer, under mild conditions. Our result contrasts with Neural Tangent Kernel and mean field analysis that assume infinite width, and contrasts with statistical mechanical approaches that assume infinite input dimension. To our best knowledge, this is a novel result.

Although we believe our paper is one step further towards understanding generalization, our paper doesn't directly address generalization, but more on student specialization. This causes some confusion from the reviewers. We sincerely apologize and we also change the title to reflect the content more accurately.

2. In the update paper, we follow R2's advice and extensively use the term "specialization" throughout the paper and clearly referenced and compared with related papers in terms of their settings. We have made clear distinction between "over-parameterization" (#parameters > #samples) versus "over-realization" (student is wider than the teacher) to reduce possible confusion.

3. We have added more explanation in the paper to explain the assumptions (in particular the small gradient assumption) and the symbols in the paragraphs.

Our assumption on small gradient at every data point is realizable due to over-realization setting. R3 suggests a more relaxed assumption: the expected gradient (over batches) is small (i.e., $E[|g|_\infty] \le \epsilon$). This in fact leads to our assumption with high probability (by Markov's inequality $P[|g|_\infty \le c \epsilon] \ge 1 - \epsilon / c$). If batch gradient is small (or zero) at every training batch, then we can derive that the gradient at every data point is also small (Theorem 1, Appendix Section 8.2).

We want to emphasize that the critical point analysis doesn't depend on weight/gradient history, as long as at $|g|_\infty \le \epsilon$ at *some* iteration, it can be applied. So the same theorem applies in expectation or in high-probability case.

4. We have rewritten the experiment section and made the setting very clear. Hopefully this address reviewers' comments. We give a clear explanation about what kind of random data we use, and how the teacher/student network is constructed. We also perform experiments when each dimension of the input following Uniform[-15,15] (Fig. 12 in Appendix Section 8.17), and show student specialization happens as well for 4 layer deep ReLU networks.

Best,
Authors.

---

### Decision · Program_Chairs · 2019-12-19

**Decision:**

Reject

**Comment:**

The article studies a student-teacher setting with over-realised student ReLU networks, with results on the types of solutions and dynamics. The reviewers found the line of work interesting, but they also raised concerns about the novelty of the presented results, the description of previous works, settings and claims, and experiments. The revision clarified some of the definitions, the nature of the observations, experiments, and related works, including a change of the title. However, the reviewers still were not convinced, in particular with the interpretation of the results, and keep their original ratings. With many points that were raised in the original reviews, the article would benefit from a more thorough revision.